# Goal-Conditioned Data Augmentation for Offline Reinforcement Learning

**Xingshuai Huang**  *xingshuai.huang@mail.mcgill.ca*
*Department of Electrical and Computer Engineering*
*McGill University*

**Di Wu**  *di.wu5@mcgill.ca*
*Department of Electrical and Computer Engineering*
*McGill University*

**Benoit Boulet**  *benoit.boulet@mcgill.ca*
*Department of Electrical and Computer Engineering*
*McGill University*

**Reviewed on OpenReview:** *https://openreview.net/forum?id=8K16dplpEO*

## Abstract

Offline reinforcement learning (RL) enables policy learning from pre-collected offline datasets, relaxing the need to interact directly with the environment. However, limited by the quality of offline datasets, it generally fails to learn well-qualified policies in suboptimal datasets. To address datasets with insufficient optimal demonstrations, we introduce ***G**oal-c**O**nditioned **D**ata **A**ugmentation* (GODA), a novel goal-conditioned diffusion-based method for augmenting samples with higher quality. Leveraging recent advancements in generative modelling, GODA incorporates a novel return-oriented goal condition with various selection mechanisms. Specifically, we introduce a controllable scaling technique to provide enhanced return-based guidance during data sampling. GODA learns a comprehensive distribution representation of the original offline datasets while generating new data with selectively higher-return goals, thereby maximizing the utility of limited optimal demonstrations. Furthermore, we propose a novel adaptive gated conditioning method for processing noisy inputs and conditions, enhancing the capture of goal-oriented guidance. We conduct experiments on the D4RL benchmark and real-world challenges, specifically traffic signal control (TSC) tasks, to demonstrate GODA's effectiveness in enhancing data quality and superior performance compared to state-of-the-art data augmentation methods across various offline RL algorithms.

## 1 Introduction

Reinforcement learning (Sutton & Barto, 2018) aims to learn a control policy from trial and error through interacting with the environment. While demonstrating remarkable performance in various domains, this approach typically requires vast amounts of training data collected from these interactions. Such data-intensive requirements become impractical in applications where environmental interactions are costly, risky, or time-consuming, such as robotics (Bhateja et al., 2024; Tang et al., 2025), autonomous driving (Li et al., 2024; Taghavifar et al., 2025), and TSC (Zhai et al., 2025; Du et al., 2024). Offline RL offers a feasible solution to these challenges by enabling policy learning directly from pre-collected historical datasets, thus significantly reducing the need to interact directly with the environment.

Although offline RL makes policy learning less expensive, its performance is highly dependent on the quality of the pre-collected datasets and may suffer from a lack of diversity, behavior policy bias, distributional shift, and suboptimal demonstrations (Prudencio et al., 2023). The performance of offline RL tends to decline

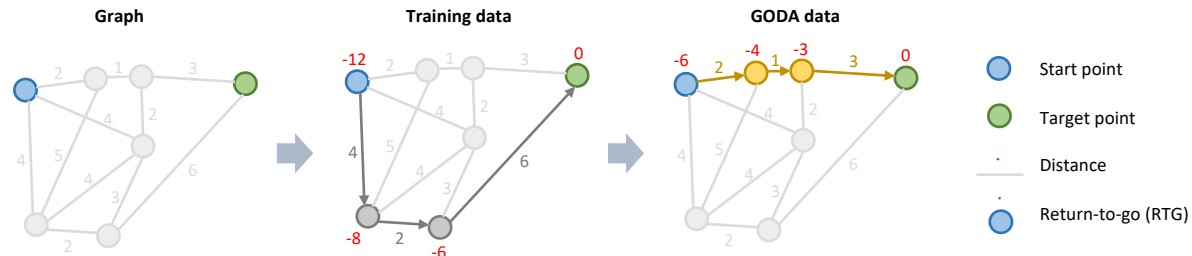

Figure 1: Illustrative examples of how GODA augments higher-return data with goal guidance. GODA utilizes scalable RTG-based goal conditions to generate samples with higher returns (shorter overall distance).

drastically when trained with suboptimal offline datasets. Previous studies have attempted to address these issues by constraining the learned policy to align closely with the behavior policy (Lyu et al., 2022) or by limiting out-of-distribution action values (Kostrikov et al., 2021). Although these approaches have shown performance improvements, they retain the inherent defects of offline datasets, remaining highly dependent on data quality.

Several studies have addressed the limitations of offline RL using data augmentation methods to generate more diverse samples. One approach involves learning world models to mimic environmental dynamics and iteratively generate synthetic rollouts from a start state (Zhang et al., 2023; Treven et al., 2024). While this method significantly improves sample efficiency and data diversity, it suffers from compounding errors and fails to control the quality of generated trajectories. Other research leverages generative models to capture the distributions of collected datasets and randomly sample new transition data (Lu et al., 2024). Although these methods demonstrate some performance improvements, they remain inefficient when dealing with datasets containing limited optimal demonstrations. This inefficiency stems from their inability to effectively control the quality of generated data.

We attempt to address this challenge by taking advantage of generative modelling to augment higher-quality data with directional goals. Unlike previous studies (Lu et al., 2024) that sample data unconditionally and randomly, we introduce GODA to incorporate representative goals, guiding the samples toward higher returns. Given the exceptional performance of diffusion models (Ho et al., 2020; Karras et al., 2022) in the field of generative artificial intelligence, GODA utilizes a diffusion model as its generative framework. GODA is trained to capture a comprehensive representation of the data distribution from the original dataset while sampling new data conditioned on selective high-return goals. This approach maximizes the utility of the limited well-performed trajectories in the original datasets.

Inspired by Decision Transformer (Chen et al., 2021), we define the 'goal' as the *return-to-go* (RTG), which represents the cumulative rewards from the current step until the end, coupled with its specific timestep in trajectories. RTG explicitly indicates the expected future rewards for a given behavior at the current timestep for a specific trajectory. We assume that at the same timestep across different trajectories, a higher RTG signifies a higher goal. To generate samples that exceed the quality of the original dataset, we introduce three *goal selection mechanisms* and a *scaling technique* to control our expected goals during sampling.

We present an illustrative example of how GODA operates in Figure 1. The task is to identify the shortest path from the starting point to the target. By setting higher RTG goals during sampling, GODA can potentially discover a more efficient route that yields a higher return (the RTG at the first timestep is equal to the return).

To better incorporate goal conditions, we further propose a novel *adaptive gated conditioning* approach. This method utilizes a condition-adaptive gated residual connection and an adaptive gated long skip connection to selectively capture multi-granularity information effectively with the guidance of goals. GODA is an off-the-shelf solution that can seamlessly integrate with other offline RL optimization approaches on various tasks to achieve superior results. We summarize our contributions:

**1)** We propose a goal-conditioned data augmentation method, namely GODA, for offline RL. It achieved enhanced data diversity and quality for offline datasets with limited optimal demonstrations.

**2)** We introduce novel *directional goals* with *selection mechanisms* and *controllable scaling* to provide higher-return guidance for the data sampling process in our employed generative models. Additionally, we propose a novel *adaptive gated conditioning approach* to better capture input information based on goal guidance.

**3)** We show GODA's competence through comprehensive experiments on the *D4RL benchmark* compared with state-of-the-art data augmentation methods across multiple offline RL algorithms. We further evaluate the effectiveness of GODA on a real-world application, i.e., *traffic signal control*, with small-size datasets obtained from widely used controllers in real-world deployments. These evaluations verify GODA's effectiveness in addressing various challenges, significantly enhancing the applicability of RL-based methods for real-world scenarios.

The remaining sections are organized as follows: Some preliminaries about offline RL and diffusion models are introduced in Section 2, followed by details about our methodology in Section 3. Section 4 describes the experimental settings about D4RL and TSC tasks, baselines and evaluation algorithms. Next, we show the data quality measurement for our augmented datasets in Section 5 and provide more detailed experimental results in Section 6. Finally, we conclude our research and discuss future directions in Section 7. The Related Work section is in Appendix A.

## 2 Preliminaries

### 2.1 Offline Reinforcement Learning

In RL, the task environment is generally formulated as a Markov decision process (MDP) $\{\mathcal{S}, \mathcal{A}, \mathcal{R}, \mathcal{P}, \gamma\}$ (Sutton & Barto, 2018). $s \in \mathcal{S}$, $s' \in \mathcal{S}$, $a \in \mathcal{A}$, $r = \mathcal{R}(s, a)$, $\mathcal{P}(s'|s, a)$, and $\gamma \in [0, 1)$ represent state, next state, action, reward function, state transition, and discount factor, respectively. RL aims to train an agent to interact with the environment and learn a policy $\pi$ from experience. The objective of RL is to maximize the expected discounted cumulative rewards over time:

$$J = \mathbb{E}_\pi \left[ \sum_{t=0}^{\infty} \gamma^t \mathcal{R}\left(s_t, a_t\right) \right], \tag{1}$$

where $t$ denotes the timestep in a trajectory. For offline RL, the policy is learned directly from offline datasets pre-collected by other behavior policies, instead of environmental interactions. The offline dataset typically consists of historical experience described as tuples $(s, a, r, s')$ and other environmental signals. After learning a policy $\pi(\mathcal{D})$ from dataset $\mathcal{D}$, the performance is evaluated in online environment as $\mathbb{E}_{\pi(\mathcal{D})} \left[ \sum_{t=0}^{\infty} \gamma^t \mathcal{R}\left(s_t, a_t\right) \right]$.

While offline RL eliminates reliance on interacting with the environment, it is highly restricted by the quality of offline datasets due to the lack of feedback from the environment. Our GODA aims to enhance the diversity and quality of the dataset by upsampling the pre-collected data to an augmented dataset $\mathcal{D}^*$. The objective is to learn a policy $\pi(\mathcal{D}^*)$ that outperform $\pi(\mathcal{D})$ learned from original dataset $\mathcal{D}$, such that

$$\mathbb{E}_{\pi(\mathcal{D}^*)} \left[ \sum_{t=0}^{\infty} \gamma^t \mathcal{R}\left(s_t, a_t\right) \right] > \mathbb{E}_{\pi(\mathcal{D})} \left[ \sum_{t=0}^{\infty} \gamma^t \mathcal{R}\left(s_t, a_t\right) \right]. \tag{2}$$

### 2.2 Diffusion Models

Diffusion models (Sohl-Dickstein et al., 2015; Ho et al., 2020; Karras et al., 2022), a class of well-known generative modeling methods, aim to learn a comprehensive representation of the data distribution $p_{\text{data}}(\mathbf{x}^N)$ with a standard deviation $\sigma_{\text{data}}$ from a given dataset. Diffusion models generally have two primary processes, the *forward process*, also known as the *diffusion process*, and the *reverse/sampling process*.

The forward process is characterized by a Markov chain in which the original data distribution $\mathbf{x}^N \in p_{\text{data}}(\mathbf{x}^N)$ is progressively perturbed with a predefined i.i.d. Gaussian noise schedule $\sigma^N = 0 < \sigma^{N-1} <$

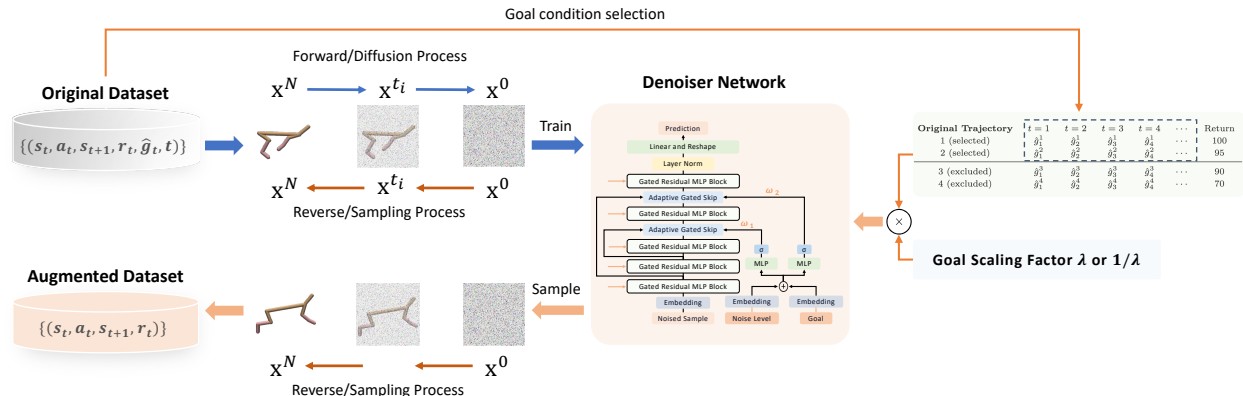

Figure 2: Illustration of diffusion model training and data augmentation. The diffusion model is first trained on the original dataset to learn the underlying data distribution, and is then used to generate higher-quality samples conditioned on selected controllable goals.

$\cdots < \sigma^0 = \sigma_{\max}$. Therefore, we can obtain a sequence of noised distributions $p(\mathbf{x}^i; \sigma^i)$ for each nose level $\sigma^i$, where the last noised distribution $p(\mathbf{x}^0; \sigma_{\max})$ can be seen as pure Guassion noise when $\sigma_{\max} \gg \sigma_{\text{data}}$.

For reverse process, Denoising Diffusion Probabilistic Model (DDPM) (Ho et al., 2020) models it as a Markov chain that involves denoising an initial noise $p(\mathbf{x}^0) = \mathcal{N}(\mathbf{x}^0; \mathbf{0}, \mathbf{I})$ to the original data distribution with learned Gaussian transitions. Elucidated Diffusion Model (EDM) (Karras et al., 2022) formulates the forward and reverse processes as a probability-flow ordinary differential equation (ODE), where the noise level can be increased or decreased by moving the ODE forward or backward in time:

$$d\mathbf{x} = -\dot{\sigma}(t_i)\sigma(t_i)\nabla_{\mathbf{x}} \log p(\mathbf{x}; \sigma(t_i))dt_i, \tag{3}$$

where $\dot{\sigma}(t_i)$ denotes derivative over denoise time and $\nabla_{\mathbf{x}} \log p(\mathbf{x}; \sigma(t_i))$ is referred to as the score function (Song et al., 2020), which points towards regions of higher data density. It is worth noting that we use $t_i$ to denote the noise time to distinguish it from the trajectory timestep $t$. The ODE pushes the samples away from the data or closer to the data through infinitesimal forward or backward steps. The corresponding step sequence is $\{t_0, t_1, ...t_N\}$, where $t_N = 0$ and $N$ denotes the number of ODE solver iterations.

EDM proposes to estimate the score function using denoising score matching (Karras et al., 2022). Specifically, a denoiser neural network $D_\theta(\mathbf{x}; \sigma)$ is trained to approximate data $\mathbf{x}^N$ sampled from $p_{\text{data}}$ by minimizing the $L_2$ denoising loss independently for each $\sigma$:

$$\min_\theta \mathbb{E}_{\mathbf{x}^N \sim p_{\text{data}}; \mathbf{n} \sim \mathcal{N}(\mathbf{0}, \sigma^2\mathbf{I})} \left\| D_\theta(\mathbf{x}^N + \mathbf{n}; \sigma) - \mathbf{x}^N \right\|_2^2. \tag{4}$$

Subsequently, the score function can be calculated as $\nabla_{\mathbf{x}} \log p(\mathbf{x}; \sigma) = (D_\theta(\mathbf{x}; \sigma) - \mathbf{x})/\sigma^2$. EDM employs Heun's $2^{\text{nd}}$ order ODE solver (Ascher & Petzold, 1998) to solve the reverse-time ODE by iteratively refining $\mathbf{x}^0$ toward $t_N = 0$, i.e., sampling data with the reverse process.

## 3 Methodology

In this section, we introduce GODA, a goal-conditioned data augmentation method utilizing generative modeling for augmenting higher-quality synthetic transition data. Our adopted diffusion model first learns comprehensive data distribution from the initial offline dataset, subsequently sampling new data towards higher return with controllable, selective goal conditions, as shown in Figure 2. In this part, we define a representative goal for GODA in Section 3.1 and introduce different selective mechanisms for goal conditions in Section 3.2. To proactively control the sampling direction, a controllable goal scaling factor is introduced in Section 3.3. For better integrating goal conditions as guidance within the diffusion model, we further propose a novel adaptive gated conditioning approach (Section 3.4) that introduces a condition-adaptive gate

mechanism into the long skip connection and the residual connection. Section 3.5 presents the pseudocode (Algorithm 1) and outlines the implementation details of our method.

## 3.1 Return-oriented goal

Prior diffusion-based data augmentation work (Lu et al., 2024) lacks the ability to guide the sampling process toward a desired outcome. In our approach, we introduce a return-oriented goal as a conditioning signal for the diffusion model, analogous to how prompts are used to guide image generation in some diffusion models (Peebles & Xie, 2023; Li et al., 2025). As shown in Figure 3, the denoiser neural network not only takes as input the noised sample and noise level, as described in Equation 4, but also an extra goal condition. Inspired by Decision Transformer, we adopt return-to-go (RTG) (Chen et al., 2021), which represents the cumulative rewards from the current step till the end, as an explicit goal condition

$$\hat{g}_t = \sum_{t'=t}^{T} r_{t'}. \tag{5}$$

For each transition sample represented as a tuple $(s, a, s', r)$ within a trajectory, the RTG serves as an unbiased measure of the future rewards corresponding to the current state-action pair. In the context of Decision Transformer, a higher RTG at a given timestep implies a more ambitious goal for the policy to achieve, thereby guiding the behavior of the agent toward more rewarding outcomes. Since the same behavior at different timesteps often yields varying RTGs across different trajectories, we combine the RTG with its corresponding timestep in the trajectory as the condition for each specific transition sample. The timestep signal acts as a timestamp for each goal.

## 3.2 Selective goal conditions

To get the training goal conditions from the original dataset, we first organize offline samples into trajectories, compute the RTG $\hat{g}_t$ for each, and append timestep $t$ to every sample. These goals along with the transition samples $\{(\hat{g}_t, t, s_t, a_t, s_{t+1}, r_t)\}$ are then used to train the diffusion model to capture the underlying data distribution. For data augmentation (sampling procedure), however, we aim to generate new samples conditioned on a selectively chosen batch of goals. To fully leverage well-performing samples and augment samples with higher returns, we propose three distinct condition selection mechanisms: return-prior, RTG-prior, and random goal conditions.

**(1) Return-prior goal condition**. In this approach, we rank all trajectories based on their return values and select the top $n$ trajectories. During the sampling procedure of the diffusion model, the RTG and timestep pairs $(\hat{g}_t, t)$ from these top $n$ trajectories are selected as the sampling goal conditions. This method filters high-return trajectories from the initial offline datasets and reuses them to sample more well-optimized transitions. The following example illustrates how we select goal conditions from the top two trajectories. $\hat{g}_t^{\tau}$ denotes the RTG value at timestep $t$ from trajectory $\tau$.

| Original Trajectory | $t = 1$ | $t = 2$ | $t = 3$ | $t = 4$ | $\cdots$ | Return |
|---|---|---|---|---|---|---|
| 1 (selected) | $\hat{g}_1^1$ | $\hat{g}_2^1$ | $\hat{g}_3^1$ | $\hat{g}_4^1$ | $\cdots$ | 100 |
| 2 (selected) | $\hat{g}_1^2$ | $\hat{g}_2^2$ | $\hat{g}_3^2$ | $\hat{g}_4^2$ | $\cdots$ | 95 |
| 3 (excluded) | $\hat{g}_1^3$ | $\hat{g}_2^3$ | $\hat{g}_3^3$ | $\hat{g}_4^3$ | $\cdots$ | 90 |
| 4 (excluded) | $\hat{g}_1^4$ | $\hat{g}_2^4$ | $\hat{g}_3^4$ | $\hat{g}_4^4$ | $\cdots$ | 70 |

**(2) RTG-prior goal condition**. We group RTGs by their associated timesteps and then sort them to select the top $n$ RTGs along with their corresponding timesteps as goal conditions. As illustrated in the example below, for each column, the RTG values are sorted in descending order from top to bottom. This approach selectively reuses high-RTG transitions for data augmentation, focusing on transitions that are most likely to yield higher returns.

**(3) Random goal condition**. We randomly select $m$ RTG and timestep pairs $(\hat{g}_t, t)$ as sampling goal conditions for each batch of samples. This increases the diversity of the augmented data while paying less attention to the optimal trajectories for improving performance.

| Re-ordered Trajectory | $t = 1$ | $t = 2$ | $t = 3$ | $t = 4$ | $\cdots$ |
|---|---|---|---|---|---|
| 1 (selected) | $\hat{g}_1^1$ | $\hat{g}_2^1$ | $\hat{g}_3^2$ | $\hat{g}_4^4$ | $\cdots$ |
| 2 (selected) | $\hat{g}_1^2$ | $\hat{g}_2^3$ | $\hat{g}_3^1$ | $\hat{g}_4^3$ | $\cdots$ |
| 3 (excluded) | $\hat{g}_1^3$ | $\hat{g}_2^2$ | $\hat{g}_3^3$ | $\hat{g}_4^1$ | $\cdots$ |
| 4 (excluded) | $\hat{g}_1^4$ | $\hat{g}_2^4$ | $\hat{g}_3^4$ | $\hat{g}_4^2$ | $\cdots$ |

## 3.3 Controllable goal scaling

Selective goal conditions offer high-return guidance during the sampling process, but are limited in generating data with returns or quality beyond the initial offline datasets. To overcome this limitation, we introduce a controllable goal scaling factor, $\lambda$, which can be multiplied with the goal values to represent a higher return expectation. This approach enables flexible adjustment of goal values to drive the sampling process toward higher-quality data. As illustrated in Figure 1, a higher RTG goal at each timestep directs the sampling process toward a trajectory with a greater overall return. Since RTG values can be either positive or negative in certain tasks, we propose multiplying positive goals by the scaling factor and dividing negative goals by it.

$$\text{goal} = \begin{cases} (\lambda \hat{g}_t, t), & \hat{g}_t >= 0 \\ (\hat{g}_t / \lambda, t), & \hat{g}_t < 0. \end{cases} \tag{6}$$

It is worth noting that we use a goal scaling factor rather than an increment factor, i.e., $\text{goal} = (\hat{g}_t + \lambda, t)$. The reason is that incrementing each RTG $\hat{g}_t = \sum_{t'=t}^{T} r_{t'}$ with a fixed value would be the same as simply incrementing the terminal reward $r_T$ by $\lambda$ while leaving all other target rewards $r_t$ for $t \neq T$ unchanged. However, assigning a higher RTG goal at each timestep is intended to encourage the agent to generate trajectories that yield higher rewards throughout the trajectory, rather than only improving the terminal reward. Therefore, a goal scaling factor can be a better choice than an increment factor.

## 3.4 Adaptive Gated Conditioning

To better capture goal guidance and seamlessly integrate conditions into the diffusion model, we propose a novel adaptive gated conditioning approach, as shown in Figure 3. This structure significantly enhances the ability to guide the diffusion and sampling processes using goal conditions. The conditional inputs include both the noise level condition and goal condition, which are embedded separately, then element-wise added, and fed into the neural network. The noise input is processed with several gated residual multi-layer perception (MLP) blocks with novel adaptive gated skip connections between shallow and deep layers.

### 3.4.1 Condition Embedding

The noise level $\sigma$ for diffusion is encoded using Random Fourier Feature (Rahimi & Recht, 2007) embedding. The RTG is processed with a linear transformation to get a hidden representation. The timestep of each RTG is embedded with Sinusoidal positional embedding (Vaswani et al., 2017). We concatenate the RTG and timestep embeddings to form the goal condition, which is then element-wise added to the noise level embedding and used as the conditional input.

### 3.4.2 Adaptive Gated Long Skip Connection

As shown in the left part of Figure 3, we adopt a long skip connection similar to U-Net (Ronneberger et al., 2015) to connect MLP blocks at different levels. To capture different information with varying importance weights, we propose a novel adaptive gated long skip connection structure by adding the previous information with an adaptive gate mechanism based on the given conditions.

$$x_{\text{out}} = (1 - \omega) * x_{\text{skip}} + \omega * x, \tag{7}$$

where $x_{\text{skip}}$ and $x$ are outputs of a shallower block and the previous block, and $\omega$ denotes a learnable weight calculated by regressing the conditional input with an MLP and a sigmoid layer.

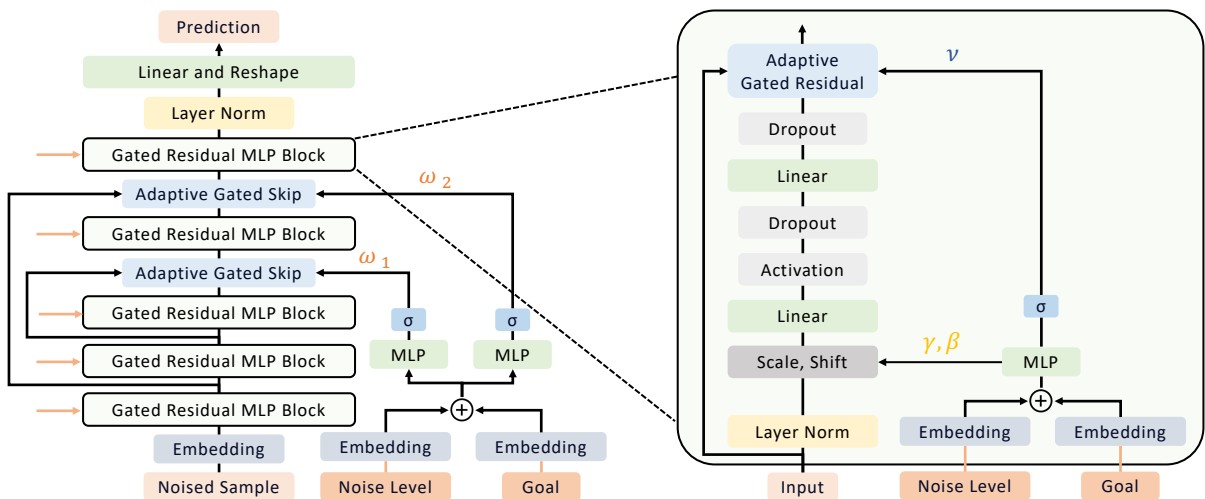

Figure 3: The denoiser neural network with adaptive gated conditioning architecture. The *condition-adaptive gated long skip connection*, shown in the left part, captures both low- and high-level features, assigning varying importance weights to each. The orange arrows to the left of each Gated Residual MLP Block represent the noise level and goal condition. The *adaptive gated residual*, depicted in the right part, further enhances the model by selectively preserving input information based on the given conditions.

### 3.4.3 Gated Residual MLP Block

The right part of Figure 3 depicts the structure of each gated residual MLP block. We mainly use a simple MLP structure to process the inputs, given the non-sequence input format. We have also tried more complicated structures, e.g., self-attention and cross-attention mechanisms, while getting degraded performance. We adopt the widely used adaptive layer normalization (adaLN) method (Peebles & Xie, 2023) to learn dimension-wise scale $\gamma$ and shift $\beta$ based on the conditional information. Besides, we explore a modification of the residual connection (He et al., 2016) and introduce a novel condition-adaptive gated residual connection to further enhance the model by selectively preserving input information. It also regresses the conditional input and gets a learnable weight $\nu$ for adaptively preserving input information.

$$x_{\text{out}} = (1 - \nu) * F(x) + \nu * x, \tag{8}$$

where $F$ is the learned transformation.

### 3.5 Method Implementation

Given the strong ability of diffusion models to capture the complex data distribution and generate high-dimensional data, we adopt EDM (Karras et al., 2022) as our generative model for augmenting offline data. The neural network equipped with adaptive gated conditioning, as illustrated in Figure 3, is used as the denoiser function. It takes as input the noised sample $\mathbf{x} + \mathbf{n}$, noise level $\sigma$, and goal condition $\mathbf{c}$, and make predictions for the original sample $\mathbf{x}$. Therefore, with goal conditions $\mathbf{c}$ and transition data $\mathbf{x}$ from the original datasets, the diffusion model $G_\theta$ with a learnable denoiser neural network $D_\theta$ is trained by

$$\min_\theta \mathbb{E}_{\mathbf{x},\mathbf{c}\sim p_{\text{data}};\mathbf{n}\sim\mathcal{N}(\mathbf{0},\sigma^2\mathbf{I})} \left\| D_\theta(\mathbf{x} + \mathbf{n}; \sigma; \mathbf{c}) - \mathbf{x} \right\|_2^2. \tag{9}$$

Algorithm 1 outlines the overall process of our GODA method. We begin by training a conditional diffusion model to approximate the data distribution of the offline dataset (Lines 2-5), using transition tuples paired with their corresponding goal conditions as training samples. Once the diffusion model is well trained, we use it to generate new transition samples by conditioning it on selected and scaled goals (Lines 6-11). The resulting augmented samples are then stored in the augmentation dataset $\mathcal{D}^*$ for subsequent policy training (Line 12). The learned policy is finally evaluated on target tasks (Line 13).

---

**Algorithm 1** GODA: Goal-Conditioned Data Augmentation

---

1: Initialize generative model $G_\theta$ and $\mathcal{D}^* = \emptyset$
   `# Diffusion Model Training`
2: Split initial offline dataset $\mathcal{D}$ into trajectories according to episode terminal information
3: Calculate RTGs for each transition sample in trajectories
4: Add RTGs and timesteps as goals into $\mathcal{D}$
5: Train $G_\theta$ on $\mathcal{D}$ using Eq. 9 by conditioning on goals
   `# Data Augmentation (Sampling)`
6: **repeat**
7:     Extract goal conditions for sampling according to the goal selection mechanism.
8:     Re-assign sampling goal conditions with goal scaling factor $\lambda$ using Eq. 6
9:     Sampling a batch of new transition samples $\mathcal{B}^*$ by conditioning on scaled goals
10:     $\mathcal{D}^* \leftarrow \mathcal{D}^* \cup \mathcal{B}^*$
11: **until** $|\mathcal{D}^*|$ = number of target samples
    `# Policy Training and Evaluation`
12: Train policy $\pi$ on the final dataset $\mathcal{D}^* \cup \mathcal{D}$
13: Evaluate policy on target tasks

---

## 4 Experimental Settings

In this section, we provide a comprehensive overview of the experiments conducted to evaluate the performance of our proposed GODA method.

### 4.1 Tasks and Datasets

#### 4.1.1 D4RL Benchmark

We adopt three popular Mujoco locomotion tasks from Gym [1], i.e., HalfCheetah, Hopper, and Walker2D, and a navigation task, i.e., Maze2D (Fu et al., 2020), as well as more complex tasks, specifically the Pen and Door tasks from the Adroit benchmark (Rajeswaran et al., 2017; Fu et al., 2020). For locomotion tasks, we adopt four data quality levels: Random, Medium-Replay, Medium, and Medium-Expert. For Maze2D, three datasets collected from different maze layouts are adopted, i.e., Umaze, Medium, and Large. For the Adroit benchmark, we use two different datasets: Human and Cloned.

#### 4.1.2 V-D4RL Benchmark

We further extend our approach to the Cheetah-Run task from V-D4RL Lu et al. (2023), a high-dimensional pixel-based offline RL benchmark, with four data quality levels: Random, Medium-Replay, Medium, and Medium-Expert. We implement our GODA by following the same settings adopted in SynthER Lu et al. (2024). Specifically, the original image-based datasets are first used to pre-train a Behavior Cloning (BC) policy network composed of a CNN encoder, a trunk network, and a final fully connected layer. The CNN encoder and trunk network are then frozen and used to transform each observation and next observation, represented as raw image data of size $84 \times 84 \times 3$, into a 50-dimensional latent representation. These low-dimensional representation data, combined with rewards and actions, are then used to train the diffusion model and augment transition samples in the latent space. These augmented samples are subsequently used to fine-tune only the final fully connected layer (output head), thereby avoiding the need to directly generate high-dimensional image data.

---

[1] https://www.gymlibrary.dev/environments/mujoco/

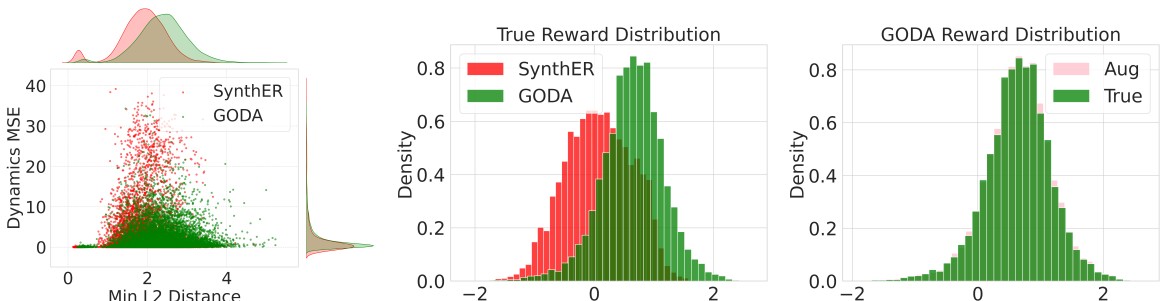

Figure 4: Data quality evaluation for SynthER and GODA on Walker2D-Random-V2. **Left**: Dynamics MSE and Min L2 Distance comparison. Smaller Dynamics MSE indicates better validity, and larger Min L2 Distance indicates higher diversity. **Middle**: Ground-truth reward distributions from the simulator for augmented datasets. **Right**: Ground-truth and augmented reward distributions for the GODA-augmented dataset.

### 4.1.3 Traffic Signal Control

To evaluate GODA's applicability to real-world challenges, we further test it on TSC tasks with much fewer training samples using the CityFlow simulator (Zhang et al., 2019). TSC aims to optimize traffic flow by efficiently managing traffic signals to maximize overall traffic efficiency.

To evaluate our GODA, we select three real-world scenarios featuring a 12-intersection grid from Jinan (JN) city and two scenarios with a 16-intersection grid from Hangzhou (HZ) city (Zhang & Deng, 2023). These scenarios represent a variety of traffic patterns and intersection structures, allowing us to cover a wide range of traffic situations. To bridge the gap between simulation and real-world conditions, we use the widely adopted Fixed-Time (FT) controller as one of our behavior policies for generating the initial offline datasets. Additionally, we employ Advanced Max Pressure (AMP) (Zhang et al., 2022) and Advanced CoLight (ACL) (Zhang et al., 2022) to create higher-quality datasets for further evaluation. We present more details in Appendix B.1.4.

### 4.2 Baseline Data Augmentation Methods

To verify the effectiveness of our proposed GODA, we compare it with three state-of-the-art data augmentation methods:

**TATU**[2] (Zhang et al., 2023), which learns world models to generate synthetic rollouts and truncates trajectories with high accumulated uncertainty.

**SynthER**[3] (Lu et al., 2024), which employs diffusion models to unconditionally augment large amounts of new data based on the learned distribution of original datasets.

**DiffStitch**[4] (Li et al., 2024), which augments data with a diffusion model and three MLPs, and actively connects low to high-reward trajectories with stitch techniques.

### 4.3 Evaluation Algorithms

To verify the quality of datasets augmented by GODA, we follow the evaluation settings adopted in previous data augmentation studies. We train two widely-used offline RL algorithms, i.e., IQL (Kostrikov et al., 2021) and TD3+BC (Fujimoto & Gu, 2021), on datasets and evaluate the learned policy on D4RL tasks. For TSC tasks, we utilize BCQ (Fujimoto et al., 2019), CQL (Kumar et al., 2020), and DataLight (Zhang & Deng, 2023) as the evaluation algorithms. For V-D4RL tasks, the BC algorithm is adopted for evaluation.

---

[2]https://github.com/pipixiaqishi1/TATU

[3]https://github.com/conglu1997/SynthER

[4]https://github.com/guangheli12/DiffStitch

Table 1: Data quality evaluation metrics for SynthER and GODA on Walker2D tasks. Smaller Dynamics MSE, larger Min L2 Distance, and larger Average Reward indicate better quality.

| Task | Dynamics MSE | | Min L2 Distance | | Average Reward | |
|------|--------------|--|-----------------|--|----------------|--|
| | SynthER | GODA | SynthER | GODA | SynthER | GODA |
| Walker2D-Random-v2 | 2.7±5.7 | **1.9±2.9** | 1.9±0.6 | **2.4±0.6** | 0.1±0.6 | **0.6±0.5** |
| Walker2D-Medium-Replay-v2 | 0.5±1.7 | **0.4±1.1** | 1.6±0.9 | **1.7±0.7** | 2.5±1.3 | **3.5±0.9** |
| Walker2D-Medium-v2 | **0.3±1.0** | **0.3±0.8** | 0.8±0.5 | **0.8±0.6** | 3.4±1.2 | **3.7±0.9** |

It is important to note that for GODA, we train the evaluation algorithms using a mix of the original datasets and the augmented datasets, whereas for the other baseline methods, only the augmented datasets are used. This is because GODA focuses on augmenting samples from the high-reward zones, which may lead to reduced data diversity. In contrast, the baseline methods, as reported in their respective papers (Lu et al., 2024) and our experiments, exhibit degraded or similar performance when using a mix of the original and augmented datasets, as illustrated in Section C.5.

## 5 Augmented Data Quality Measurement

Since our GODA is built upon SynthER, we compare the quality of the datasets augmented by both SynthER and GODA to assess whether the goal conditions incorporated by GODA enhance data quality. We adopt two metrics, i.e., Dynamics MSE and Min L2 Distance, from SynthER (Lu et al., 2024):

$$\text{Dynamics MSE} = \frac{1}{M} \sum_{i=1}^{M} \left( (s_{t+1}^i, r_t^i) - (\hat{s}_{t+1}^i, \hat{r}_t^i) \right)^2, \tag{10}$$

$$\text{Min L2 Distance} = \frac{1}{M} \sum_{i=1}^{M} ||(s_t^i, a_t^i) - (\bar{s}_t^i, \bar{a}_t^i)||_2, \tag{11}$$

and introduce an Average Reward for evaluating reward distributions of augmented datasets

$$\text{Average Reward} = \frac{1}{M} \sum_{i=1}^{M} \hat{r}_t^i, \tag{12}$$

where $M$ is the selected number of samples, $s_t^i, a_t^i, s_{t+1}^i, r_t^i$ denote the samples from augmented datasets, $\hat{s}_{t+1}^i$ and $\hat{r}_t^i$ denote the next state and reward generated by the simulator given states and actions from augmented datasets, and $\bar{s}_t^i$ and $\bar{a}_t^i$ are the state and action from original datasets.

Dynamics MSE measures how well the augmentation models capture the dynamics of the environment by learning patterns from the original datasets and generating data that aligns with those dynamics. Min L2 Distance assesses the models' exploration capabilities and data diversity by calculating the average L2 distance between each augmented sample and its nearest neighbor in the original dataset, reflecting how diverse the generated data is. Average Reward compares the ground-truth reward distributions produced by the simulator given states and actions in datasets augmented by SynthER and GODA.

The left part of Figure 4 presents a scatter plot of 10K points sampled from the augmented datasets. Results show that datasets generated by GODA exhibit much lower Dynamics MSE and a wider range of Min L2 Distance values, indicating both better alignment with environmental dynamics and greater diversity. The superior validity in performance may stem from the goal conditions (RTG-timestep pairs), which provide critical information for generating samples that better match the environment's dynamics. Meanwhile, the increased diversity is likely due to the scaled-out-of-distribution goal conditions incorporated in the sampling process.

The middle part demonstrates that GODA not only generates samples within a high-reward data zone but also extends the boundary of high rewards beyond the best demonstrations, compared to SynthER. The right part shows that the rewards generated by GODA align closely with the ground-truth values.

Table 2: Normalized scores of D4RL Gym and Maze2D tasks for GODA and baseline data augmentation methods. The results are calculated across 5 random seeds. Values in bold represent the best performance (largest score). 'HC': HalfCheetah; 'WK': Walker2D; 'HP': Hopper; 'MZ': Maze2D.

| Task | Data | IQL Original | TATU | SynthER | DStitch | GODA | TD3+BC Original | TATU | SynthER | DStitch | GODA |
|---|---|---|---|---|---|---|---|---|---|---|---|
| HC | Rand | 15.2±1.2 | 17.7±2.9 | 17.2±3.4 | 15.8±2.0 | **19.5±0.5** | 11.3±0.8 | 12.1±2.3 | 12.2±1.1 | 11.8±1.4 | **12.5±1.3** |
|  | Med-R | 43.5±0.4 | 44.2±0.1 | 46.6±0.2 | 44.7±0.1 | **47.5±0.4** | 44.8±0.7 | 44.5±0.3 | **45.9±0.9** | 44.7±0.3 | 44.9±0.2 |
|  | Med | 48.3±0.1 | 48.2±0.1 | 49.6±0.3 | 49.4±0.1 | **50.4±0.1** | 48.1±0.2 | 48.1±0.2 | 49.9±1.2 | **50.4±0.5** | 48.5±0.1 |
|  | Med-E | **94.6±0.2** | 94.4±0.6 | 93.3±2.6 | 94.4±1.4 | 94.0±1.1 | 90.8±7.0 | 89.3±3.9 | 87.2±11.1 | **96.0±0.5** | 94.5±7.7 |
| WK | Rand | 4.1±0.8 | 6.3±0.5 | 4.2±0.3 | 4.6±1.1 | **14.3±7.1** | 0.6±0.3 | **6.5±4.3** | 2.3±1.9 | 2.4±1.0 | 4.2±1.8 |
|  | Med-R | 82.6±8.0 | 75.0±12.1 | 83.3±5.9 | 86.6±2.8 | **96.1±4.9** | 85.6±4.6 | 62.1±10.4 | 90.5±4.3 | 89.7±4.2 | **93.0±5.6** |
|  | Med | 84.0±5.4 | 76.6±10.7 | **84.7±5.5** | 83.2±2.2 | 79.1±2.4 | 82.7±5.5 | 75.8±3.5 | 84.8±1.4 | 83.4±1.7 | **86.2±0.7** |
|  | Med-E | **111.7±0.6** | 110.7±0.6 | 111.4±0.7 | 111.6±0.1 | 110.9±0.7 | 110.0±0.4 | **110.7±0.7** | 110.2±0.5 | 110.2±0.3 | **110.7±0.5** |
| HP | Rand | 7.2±0.2 | 8.1±2.9 | 7.7±0.1 | 6.5±0.9 | **8.7±2.1** | 8.6±0.3 | **18.1±11.5** | 14.6±9.4 | 8.8±2.3 | 8.2±0.1 |
|  | Med-R | 84.6±13.5 | 79.6±7.6 | **103.2±0.4** | 102.1±0.4 | 102.5±0.6 | 64.4±24.8 | 64.1±10.5 | 53.4±15.5 | **79.6±13.5** | 63.0±12.8 |
|  | Med | 62.8±6.0 | 60.3±3.6 | 72.0±4.5 | 71.0±4.2 | **74.3±2.9** | 60.4±4.0 | 58.3±4.8 | 63.4±4.2 | 60.3±4.9 | **74.8±3.6** |
|  | Med-E | 106.2±6.1 | 93.4±17.8 | 90.8±17.9 | **110.9±2.9** | 96.9±12.3 | 101.1±10.5 | 99.0±14.9 | 105.4±9.7 | 107.1±7.0 | **107.5±15.3** |
| **Average** |  | 62.1±3.5 | 59.5±5.0 | 63.7±3.5 | 65.1±1.5 | **66.2±2.9** | 59.0±4.9 | 57.4±5.6 | 60.0±5.1 | 62.0±3.1 | **62.4±4.1** |
| MZ | Umaze | 37.7±2.0 | 33.0±4.8 | 41.0±0.7 | 38.5±6.2 | **59.5±2.6** | 29.4±14.2 | 37.7±10.9 | 37.6±14.4 | 38.4±7.5 | **46.4±8.3** |
|  | Med | 35.5±1.0 | 35.1±1.3 | 35.1±2.6 | 35.5±1.5 | **35.8±2.6** | 59.5±41.9 | 73.8±36.9 | 65.2±36.1 | 66.8±30.9 | **86.5±26.4** |
|  | Large | 49.6±22.0 | 69.1±20.1 | 60.8±5.3 | 68.4±12.6 | **109±16.5** | 97.1±29.3 | 93.1±25.3 | 92.5±38.5 | 92.4±36.2 | **104.3±20.1** |
| **Average** |  | 40.9±8.3 | 45.7±8.2 | 45.6±2.9 | 47.5±6.8 | **68.1±6.6** | 62.0±28.2 | 68.2±10.6 | 65.1±29.7 | 65.9±24.9 | **79.1±7.5** |

Table 3: Normalized scores of GODA and baseline data augmentation methods on Adroit tasks evaluated using IQL.

| Task | Dataset | Original | TATU | SynthER | DStitch | GODA |
|---|---|---|---|---|---|---|
| Pen | Human | 79.1±28.5 | 88.9±22.6 | **96.8±8.6** | 87.4±8.6 | 75.6±31.4 |
|  | Cloned | 45.8±29.9 | 52.5±27.9 | 45.3±23.4 | 64.0±29.6 | **64.8±20.6** |
| **Average** |  | 62.4±29.2 | 70.7±25.2 | 71.0±16.0 | **75.7±19.1** | 70.2±26.0 |
| Door | Human | 1.6±2.1 | 7.0±1.6 | 8.3±2.2 | 10.0±2.5 | **14.8±5.0** |
|  | Cloned | -0.1±0.5 | -0.1±0.3 | 5.9±1.8 | 4.4±0.4 | **16.8±6.1** |
| **Average** |  | 0.8±1.3 | 3.5±1.0 | 7.1±2.0 | 7.2±1.4 | **15.8±5.5** |

The evaluation results for the three metrics in Table 1 further demonstrate that GODA outperforms SynthER in terms of all data quality evaluation metrics across nearly all Walker2D tasks. Notably, GODA is particularly effective on random-level datasets with high randomness and sparse optimal demonstrations, where traditional augmentation methods often fall short.

Table 4: Evaluation of GODA on the Cheetah-Run task from the pixel-based V-D4RL benchmark using the BC algorithm, with sample augmentation performed in the latent space.

| Eval Algorithm | Aug Methods | Rand | Med-R | Med | Med-E | Average |
|---|---|---|---|---|---|---|
| BC | Original | 0.0±0.0 | 25.0±3.6 | 51.6±1.4 | 57.5±6.3 | 33.5±2.8 |
|  | SynthER | 0.1±0.0 | 27.9±3.4 | 52.2±1.2 | **69.9±9.5** | 37.5±3.5 |
|  | GODA | **1.4±0.8** | **35.5±6.1** | **54.9±2.0** | 67.6±10.4 | **39.9±4.8** |

## 6 Experimental Results

After directly evaluating the data quality of the augmented datasets using various metrics, we further assess GODA by training policies on these datasets and verifying their performance on corresponding tasks. This section presents the overall performance comparison between our GODA and other baseline data augmentation methods.

### 6.1 Offline Reinforcement Learning Evaluation

#### 6.1.1 Performance on D4RL

**Gym tasks.** Table 2 presents the normalized score comparison between GODA and other state-of-the-art data augmentation methods trained on the D4RL Gym and Maze2D. We adopt the results of Original and SyntheER from the SynthER paper (Lu et al., 2024), and those of TATU and DStitch from the DStitch

Table 5: Average travel time comparison on real-world TSC tasks. Smaller travel time indicates better traffic efficiency.

| Traffic | Dataset | BCQ | | | CQL | | | DataLight | | |
|---|---|---|---|---|---|---|---|---|---|---|
| | | Original | SynthER | GODA | Original | SynthER | GODA | Original | SynthER | GODA |
| JN 1 | FT | 269.7±4.1 | 267.9±2.7 | **264.1±4.4** | 272.0±2.1 | 273.4±2.5 | **271.7±4.5** | 279.8±2.9 | 274.1±1.6 | **270.7±4.1** |
| | AMP | 271.5±3.9 | 264.0±6.1 | **259.7±0.9** | 261.8±0.3 | 261.7±4.3 | **260.6±3.4** | **298.1±3.1** | 299.2±2.0 | 301.7±1.4 |
| | ACL | 271.1±2.4 | 271.9±1.4 | **270.6±0.3** | 273.3±1.6 | 275.2±4.4 | **273.2±4.3** | 256.4±3.2 | 258.4±2.5 | **255.3±0.3** |
| JN 2 | FT | 267.2±3.6 | **265.5±1.2** | 266.6±5.1 | **269.3±0.3** | 275.3±6.9 | 272.5±1.9 | 293.9±1.7 | 288.2±2.2 | **281.0±0.3** |
| | AMP | **250.7±0.7** | 254.7±2.6 | 252.1±3.4 | 251.9±4.0 | 249.4±5.3 | **245.5±1.0** | 244.4±2.3 | **240.0±2.2** | 240.9±4.0 |
| | ACL | **253.4±0.3** | 265.7±1.5 | 262.0±4.5 | 248.1±0.3 | 248.2±2.1 | **248.0±2.0** | 241.9±0.3 | 236.4±0.3 | **235.1±0.5** |
| JN 3 | FT | 266.9±3.6 | 263.5±4.9 | **257.3±3.4** | 268.0±0.7 | 273.7±2.3 | **267.1±2.8** | 302.6±1.9 | 299.9±5.0 | **299.8±1.8** |
| | AMP | 263.8±3.1 | 259.3±0.7 | **253.2±4.4** | 251.5±2.9 | 247.7±4.9 | **242.5±3.5** | 239.4±1.8 | 241.8±4.9 | **232.5±1.7** |
| | ACL | **242.2±4.0** | 245.7±0.6 | 243.2±1.6 | **242.1±1.4** | 244.5±4.0 | 245.5±7.1 | 240.3±3.6 | 234.7±1.3 | **230.1±2.7** |
| **Average** | | 267.9±3.2 | 267.1±2.3 | **263.9±2.7** | 264.8±1.9 | 265.7±4.2 | **262.4±3.2** | 267.5±2.1 | 265.5±2.3 | **262.6±1.4** |
| HZ 1 | FT | 324.5±7.3 | 313.2±2.0 | **310.5±0.8** | 317.4±5.7 | 315.5±4.0 | **307.1±2.7** | 290.1±0.6 | 290.8±0.3 | **287.2±0.3** |
| | AMP | **295.8±4.6** | 302.7±1.9 | 301.7±4.1 | 300.0±0.3 | 295.1±0.3 | **285.4±1.0** | 287.3±4.3 | **284.9±3.8** | 286.1±3.2 |
| | ACL | 281.7±1.1 | 281.3±6.5 | **281.1±1.3** | 288.6±3.3 | 286.3±2.8 | **278.9±2.1** | 284.9±5.6 | 282.1±1.7 | **278.0±3.8** |
| HZ 2 | FT | **340.0±4.9** | 340.7±4.6 | 341.3±1.5 | 341.7±2.7 | 334.8±1.4 | **331.6±1.9** | 308.0±0.3 | **308.4±3.1** | 309.2±3.0 |
| | AMP | 332.5±0.3 | 324.8±3.0 | **316.9±1.9** | 318.0±0.6 | 321.9±3.3 | 321.4±3.7 | 312.3±3.6 | 310.4±2.2 | **308.4±2.2** |
| | ACL | 336.7±1.9 | 329.3±0.5 | **327.1±2.8** | 347.4±3.8 | 343.9±2.6 | **336.8±0.3** | 317.6±4.1 | **314.8±4.1** | 315.4±3.0 |
| **Average** | | 317.3±2.6 | 315.8±3.3 | **313.6±2.3** | 319.1±2.1 | 316.4±2.1 | **310.8±1.8** | 302.0±3.6 | 300.1±3.0 | **299.4±3.0** |

(Li et al., 2024) paper. We further conduct experiments for tasks not covered in the literature. As shown in the results, GODA consistently outperforms other methods across most Gym locomotion tasks when evaluated with both IQL and TD3+BC, resulting in higher average scores. Notably, even for tasks using Random datasets, GODA successfully leverages limited high-quality samples to enhance data quality, leading to improved final performance. GODA boosts performance on Med and Med-R datasets; however, a notable gap still remains between the performance of algorithms trained on GODA-augmented Med/Med-R datasets and the performance of algorithms trained on the original Med-E datasets. These limitations suggest that while GODA is effective at maximizing the utility of suboptimal data, it cannot fully synthesize expert-level performance from non-expert data alone. Its strength lies in extracting and amplifying high-quality signals within existing data, not inventing expert behavior from scratch.

**Maze2D tasks.** For Maze2D tasks where rewards are sparse, GODA demonstrates significant improvements across all datasets, achieving average gains of 43.4% and 16.0% over the best baseline methods when evaluated with IQL and TD3+BC, respectively. The maximum improvement reaches 57.7% when applying GODA to the Maze2D Large dataset. This might be because Maze2D tasks involve navigation from a start to a goal location, similar to the illustrative example in Figure 1, making the goal naturally explicit and spatially defined. GODA can directly condition the learning process on the desired goal position, which aligns perfectly with the task's structure. Besides, GODA can possibly reuse sub-trajectories and recombine transitions from different paths that end near the same goal, making data augmentation more effective and less likely to violate task constraints. These results highlights GODA's ability to effectively capture data distributions of various types of datasets and consistently augment high-quality samples.

**Adroit tasks.** Table 3 presents the normalized scores on Adroit Pena and Door tasks, evaluated using the IQL algorithm, as TD3+BC fails to perform on these tasks. As shown, GODA outperforms all baselines on the Pen-Cloned dataset, although it underperforms on Pen-Human. This might be because Pen task involves high-dimensional, dexterous manipulation, where small perturbations in action can lead to disproportionately large changes in reward (e.g., minor deviations in gripper position can affect pen stability). GODA's reliance on RTG as a conditioning signal can be limiting in such scenarios. However, for both datasets in the Door task, GODA demonstrates significant improvements over the best baseline methods. These results further demonstrate that GODA is capable of handling more complex tasks.

### 6.1.2 Performance on V-D4RL

Table 4 shows the results of the BC algorithm evaluated on the pixel-based Cheetah-Run task, with data augmentation performed in the latent space. The results indicate that GODA achieves the best performance on three out of four dataset quality levels and obtains the highest average score. It outperforms the BC algorithm without data augmentation by 19.1%, and the BC with SynthER-augmented samples in the latent

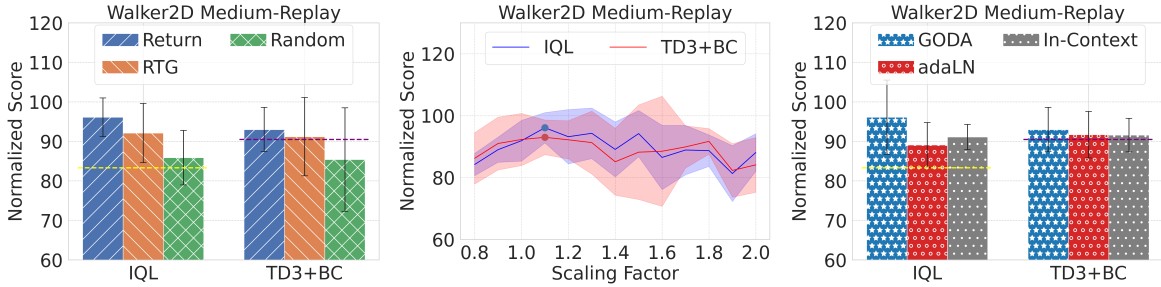

Figure 5: Ablation studies on condition selection mechanism, goal scaling factor, and adaptive gated conditioning from left to right. Yellow and purple horizon lines represent results for SynthER.

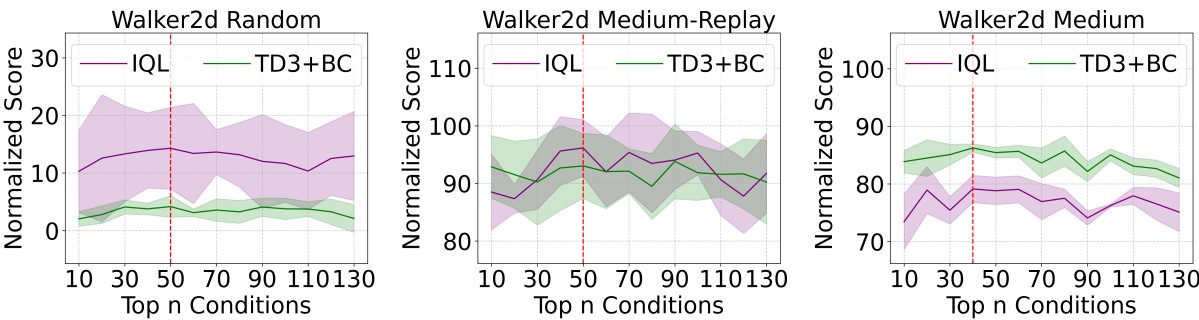

Figure 6: Ablation study on top n conditions.

space by 6.4%. Notably, GODA delivers substantial improvements on the Random and Medium-Replay datasets, highlighting its suitability for offline scenarios with limited expert demonstrations.

### 6.1.3 Performance on Traffic Signal Control

Table 5 presents a comparison of average travel times across different methods for TSC tasks. As shown, while SynthER achieves modest improvements over the performance of models trained on the original datasets, it fails to surpass the original datasets on JN tasks when using the CQL algorithm. In contrast, our GODA consistently outperforms both the original datasets and SynthER across most tasks and all average evaluations. These extended experiments on TSC tasks further validate that GODA is not only applicable to diverse tasks but also capable of improving performance in real-world challenges.

## 6.2 Ablation Study

To validate the effectiveness of GODA's components, we conduct experiments using different configurations.

### 6.2.1 Ablation on Condition Selection

We test three condition selection mechanisms as described in Section 3.2: return-prior, RTG-prior, and random goal conditions. As shown in the left part of Figure 5, the return-prior method demonstrates superior performance compared to the other two approaches. Moreover, GODA with the return- and RTG-prior conditions outperforms SynthER when tested on two offline RL algorithms. In contrast, the random-prior method shows results comparable to SynthER. This suggests that high-goal conditions identified by the return- and RTG-prior methods enable GODA to generate samples beyond the original data distribution. Randomly selected goal conditions, however, fail to target high-reward regions, producing similar results to SynthER.

Table 6: Ablation study on adaptive gated long skip connection and adaptive gated residual connection. **AdaLN** removes both the *adaptive gated long skip connection* and *adaptive gated residual connection* but only retains the AdaLN structure; **w/o AG-LSC** removes the *adaptive gated long skip connection*; **w/o AG-RC** removes the *adaptive gated residual connection*; **w/o Gate-LSC** retains the long skip connection, but without any *gating* mechanism; and **w/o Gate-RC** retains the residual connection, but without *gating*.

| Task | Eval Alg. | AdaLN | w/o AG-LSC | w/o AG-RC | w/o Gate-LSC | w/o Gate-RC | GODA |
|---|---|---|---|---|---|---|---|
| **WK2D** | IQL | 89.1±5.7 | 90.2±8.2 | 94.2±6.1 | 92.4±8.6 | 95.0±11.7 | **96.1±9.4** |
| | TD3+BC | 91.7±5.9 | 92.2±4.9 | 92.5±7.9 | 91.9±6.5 | 92.9±9.0 | **93.0±5.6** |
| **Average** | | 90.4±5.8 | 91.2±6.6 | 93.4±4.0 | 92.2±7.6 | 94.0±10.4 | **94.6±7.5** |

Moreover, return-prior generally outperforms the RTG-prior. This may be because using a return-prior aligns better with the distribution of the original trajectories. However, directly applying the RTG prior can introduce inconsistencies when reordering trajectories, as illustrated in Section 3.2. Specifically, the resulting rewards (calculated as the difference between the RTGs of the current and next timesteps) in the re-ordered trajectories may sometimes lose important high-reward signals, since RTGs are extracted from different original trajectories and may no longer stay in high-reward regions.

### 6.2.2 Ablation on Goal Scaling Factor

We further examine the effect of different scaling factors on D4RL tasks, testing values ranging from 0.8 to 2.0. As seen in the middle part of Figure 5, the performance improves as the scaling factor increases, but slightly degrades when the scaling factor exceeds 1.1. Scaling factors below 1.0 shrink the selected goals, leading to suboptimal samples. Conversely, scaling factors above 1.1 push the selected goals too far beyond the training data distribution, resulting in diminished performance. Further ablation studies can be seen in Appendix C.4.

### 6.2.3 Ablation on Adaptive Gated Conditioning

Moreover, we evaluate the impact of the adaptive gated conditioning method. We compare GODA with two variants: one using only adaLN conditioning (Peebles & Xie, 2023), and another using in-context conditioning, where condition embeddings are directly appended to the input embeddings. From the right part of Figure 5, it is clear that GODA with adaptive gated conditioning achieves the best results, and the adaLN and in-context conditioning show similar performance. Additionally, all three methods outperform SynthER, which lacks goal conditions. This demonstrates that the inclusion of goal conditions is crucial for guiding the sampling process toward high returns, and our adaptive gated conditioning method enhances the model's ability to fully utilize these conditions.

We conduct further ablation experiments on both the condition-adaptive gated long skip connection (Section 3.4.2) and condition-adaptive gated residual connection (Section 3.4.3), and each adaptive gating component as well. The results in Table 6 show that removing either the adaptive gated long skip connection or the adaptive gated residual connection leads to performance degradation, with the adaptive gated long skip connection presenting a more pronounced impact. Replacing the adaptive gated long skip connection with a vanilla long skip connection, i.e., without any adaptive gating, also results in a significant performance drop, though it still performs slightly better than having no long skip connection at all. This highlights the importance of incorporating goal information into the conditioning structure. A similar trend is observed for the adaptive gated residual connection.

### 6.2.4 Ablation on Top Conditions Selection

Given that the original datasets contain varying numbers of trajectories, the number of top conditions selected for sampling may differ across datasets. We compare different selections of the top $n$ conditions for each D4RL dataset. Based on the results shown in Figure 6, we empirically select 50 top conditions for the Random and Medium-Replay datasets, and 40 for the Medium datasets.

While this hyperparameter may seem less critical than $\lambda$, it can still meaningfully influence performance. For example, as shown in Figure 7, the performance difference between the best and worst top-$n$ settings on Walker2D Medium-Replay using IQL reaches 8.8, which is a notable gap. The top-$n$ selection primarily acts as a filter, ensuring that data augmentation focuses on high-return regions. However, increasing $n$ can introduce more diversity, allowing the diffusion model to learn from a broader range of transitions. This represents a classic trade-off between data quality and diversity: smaller $n$ prioritizes high-return samples, while larger $n$ can improve generalization through diversity.

## 7 Conclusion and Discussion

This paper proposes a novel goal-conditioned data augmentation method for offline RL, namely GODA, which integrates goal guidance into the data augmentation process. We define the easily obtainable return-to-go signal, along with its corresponding timestep in a trajectory, as the goal condition. To achieve high-return augmentation, we introduce several goal selection mechanisms and a scaling method. Additionally, we propose a novel adaptive gated conditioning structure to better incorporate goal conditions into our diffusion model. We demonstrate that data augmented by GODA shows higher quality than SynthER without goal conditions on different evaluation metrics. Extensive experiments on the D4RL benchmark confirm that GODA enhances the performance of classic offline RL methods when trained on GODA-augmented datasets. Furthermore, we evaluate GODA on real-world traffic signal control tasks. The results demonstrate that GODA is highly applicable to TSC problems, even with very small real-world training datasets, making RL-based methods more practical for real-world applications.

We acknowledge that while GODA is effective at maximizing the utility of suboptimal data, it cannot fully synthesize expert-level performance from non-expert data alone. Its strength lies in extracting and amplifying high-quality signals within existing data, not inventing expert behavior from scratch. Besides, GODA can sometimes fail to perform well on some datasets with high reward sensitivity, such as Adroit Pen-Human. Moreover, our current method is not directly applicable to high-dimensional, pixel-based visual RL or real-world robotics datasets, as it requires transforming image data into low-dimensional latent representations before applying diffusion-based data augmentation. Since our approach operates in a low-dimensional feature space, this additional transformation step can complicate the implementation and introduce dependencies on the architecture of the policy network. In future work, we aim to develop more direct methods for augmenting pixel-based datasets by improving the sampling efficiency of the diffusion model. We also plan to explore the application of GODA to more complex, real-world scenarios.

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

# A Related Work

## A.1 Offline Reinforcement Learning

Offline RL involves learning policies from pre-collected offline datasets comprising trajectory rollouts generated by behavior policies. This approach is promising because it avoids the costs and risks associated with direct interactions with the environment. Conventional offline RL methods aim to alleviate the distributional shift problem, i.e., a significant drop in performance due to deviations between the learned policy and the behavior policy used for generating the offline data (Hu et al., 2023). To address this issue, various strategies have been employed, including explicit correction (Xu et al., 2022), such as constraining the policy to a restricted action space (Kumar et al., 2019), and making conservative estimates of the value function (Yu et al., 2021; Kumar et al., 2020), with the aim of aligning the behavior policy with the learned policy.

Some recent studies exploit the strong sequence modeling ability of Transformer models to solve offline RL with trajectory optimization. For instance, Decision Transformer (Chen et al., 2021) and its variants (Wu et al., 2024; Gao et al., 2024) utilize a GPT model to autoregressively predict actions given the recent subtrajectories composed of historical RTGs, states, and actions. These approaches integrate hindsight return information, i.e., RTG, with sequence modeling, eliminating the necessity for dynamic programming.

Diffusion models have also been adopted in offline RL, given their exceptional capability of multi-modal distribution modeling. Diffuser (Janner et al., 2022) employs diffusion models for long-horizon planning, effectively bypassing the compounding errors associated with classical model-based RL. Hierarchical Diffuser (Chen et al., 2024) enhances this approach by introducing a hierarchical structure, specifically a jumpy planning method, to improve planning effectiveness further.

## A.2 Data Augmentation in Offline Reinforcement Learning

Rather than passively reusing data and concentrating on enhancing learning algorithms, data augmentation proactively generates more diverse data to improve policy optimization. Some model-based RL methods employ learned world models from offline datasets to simulate the environment and iteratively generate synthetic trajectories, facilitating policy optimization (Zhang et al., 2023). For instance, TATU (Zhang et al., 2023) uses world models to produce synthetic trajectories and truncates those with high accumulated uncertainty. However, model-based RL often suffers from compounding errors in the learned world models. Some basic data augmentation functions (DAFs), i.e., translate, rotate, and reflect, are also applied to augment trajectory segments (Pitis et al., 2020). GuDA (Corrado et al., 2024) further introduces human guidance into these DAFs to improve data quality, while human intervention is costly and lacks scalability. Diffusion models are also directly applied to data augmentation through the sampling process. SynthER (Lu et al., 2024) is the first work that employs diffusion models to learn the distribution of initial offline datasets and unconditionally augment large amounts of new random data. However, it fails to control the sampling process to steer toward high-return directions actively. DiffStitch (Li et al., 2024) attempts to enhance the quality of generated data by actively connecting low-reward trajectories to high-reward ones using a stitching technique.

We propose enhancing the quality of generated data from a different perspective by introducing a controllable directional goal into our generative modeling. This approach selectively reuses optimal trajectories to guide the sampling process toward achieving higher returns.

# B Experiment Details

## B.1 Hyperparameters

We show more details about the hyperparameter settings of the GODA model.

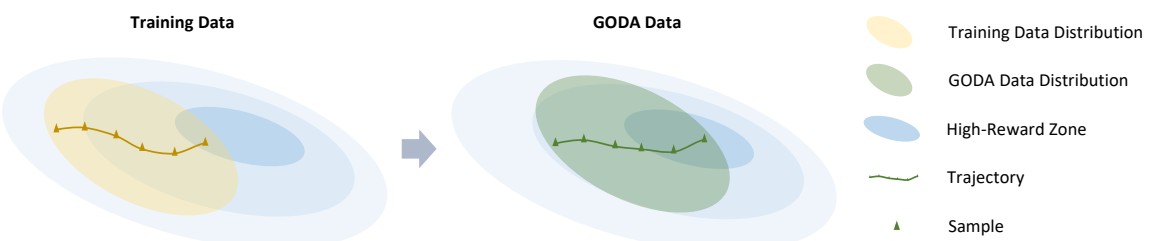

Figure 7: An illustrative example of how GODA utilizes higher goals to steer the sampling process toward a higher-reward data distribution region.

Table 7: Hyperparameters for denoiser network.

| Hyperparameter | Value |
|---|---|
| embedding dimension | 128 |
| MLP width | 512 |
| MLP activation | SiLU |
| gate activation | Sigmoid |
| learning rate | 0.0003 |
| batch size | 256 |
| learning rate schedule | cosine annealing |
| optimizer | Adam |
| gradient update steps | 100K |

### B.1.1 Denoising Network

The denoising neural network utilizes the adaptive gated conditioning architecture, as shown in Figure ref fig:gac. Table 7 details the associated hyperparameters. We use Random Fourier Feature embedding (Rahimi & Recht, 2007) and Sinusoidal positional embedding (Vaswani et al., 2017) to process the noise level and timestep of each RTG, respectively, with an embedding dimension of 128. The width of the linear layers in the MLP block is set to 512, with SiLU (Elfwing et al., 2018) as the activation function. The total number of trainable parameters for the denoiser neural network is approximately 3.3 M. We train our GODA model with 100K steps of gradient updates, with a batch size of 256 and a learning rate of 0.0003.

### B.1.2 Elucidated Diffusion Model

We adopt EDM (Karras et al., 2022) as our diffusion model and follow the original settings from SynthER (Lu et al., 2024), with the default hyperparameters shown in Table 8. EDM employs Heun's 2nd order ODE solver (Ascher & Petzold, 1998) to solve the reverse-time ODE, enabling data sampling through the reverse process. The diffusion timestep is set to 128 for higher-quality results. All training and sampling are conducted on an AMD Ryzen 7 7700X 8-Core Processor and a single NVIDIA GeForce RTX 4080 GPU. Training GODA

Table 8: Hyperparameters for the diffusion model.

| Hyperparameter | Value |
|---|---|
| number of diffusion steps | 128 |
| $S_{\text{churn}}$ | 80 |
| $S_{\text{tmin}}$ | 0.05 |
| $S_{\text{tmax}}$ | 50 |
| $S_{\text{noise}}$ | 1.003 |
| $\sigma_{\min}$ | 0.002 |
| $\sigma_{\max}$ | 80 |

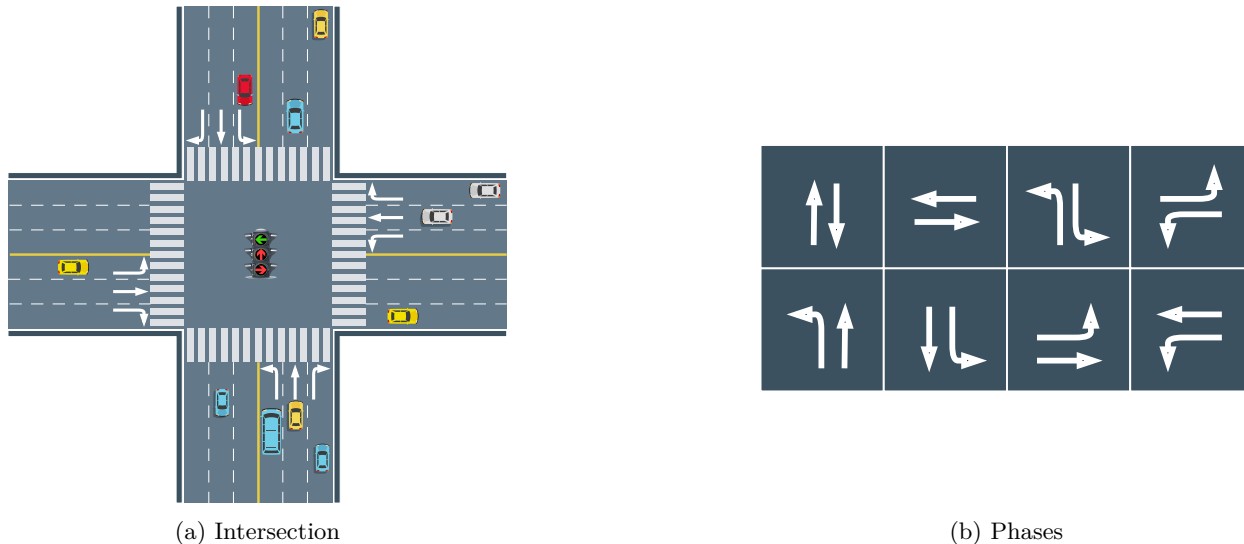

(a) Intersection                                         (b) Phases

Figure 8: A standard signalized intersection with four 3-lane approaches. The right part shows eight different phase settings.

for 100K steps takes approximately 14 minutes, while generating 5M samples with a sampling batch size of 250K requires about 300 seconds.

### B.1.3 D4RL Tasks

In the case of **Gym** tasks where dense rewards are available, we employ three distinct data configurations from the D4RL datasets: Random, Medium-Replay, and Medium. To elaborate, Random datasets contain transition data generated by a randomly initialized policy. Medium datasets consist of a million data points gathered using a policy that achieves one-third of the performance of an expert approach. Medium-replay datasets contain the stored experience in a replay buffer during the training of a policy until it reaches the score in Medium datasets. Medium-Expert datasets consist of a 50/50 mixture of expert demonstrations and suboptimal trajectories.

For **Maze2D**, a 2D agent is trained to reach a goal position utilizing minimal feedback, i.e., a single point for success, zero otherwise. Three datasets collected from different maze layouts are adopted, i.e., Umaze, Medium, and Large.

The Pen and Door tasks from the **Adroit** benchmark (Rajeswaran et al., 2017; Fu et al., 2020) involve manipulating a pen and opening a door using a 24-DoF simulated hand robot. For each task, we use two different datasets: Human and Cloned. The Human dataset consists of trajectories from human demonstrations, while the Cloned dataset is generated by applying an imitation policy trained from a mix of human and expert demonstrations and combining the resulting trajectories with the demonstrations in a 50/50 split.

We augment 5M samples for each D4RL task. For the sampling process, we use the return-prior goal condition selection method and set the goal scaling factor to 1.1 for all tasks. We use the implementation of IQL and TD3+BC from the Clean Offline Reinforcement Learning (CORL) codebase (Tarasov et al., 2024) for D4RL tasks.

### B.1.4 TSC Tasks

As shown in Figure 8, a signalized intersection in TSC problems is composed of approaches with several lanes in each approach. The controller manages the phase as shown in the top right part of Figure 8, which determines the activated traffic signals for different directions, to control the orderly movement of vehicles.

We formulate the TSC problem as an MDP and define the state, action, and the reward function as follows:

Table 9: Performance comparison on Gym tasks using DMG as the evaluation algorithm. Values in bold represent the best performance (**largest** normalized score).

| Task | Data | DMG | | |
| | | Original | SynthER | GODA |
|---|---|---|---|---|
| **HalfCheetah-v2** | Random | 27.3±2.5 | 27.8±1.9 | **31.5±4.3** |
| | Medium-Replay | **51.4±1.6** | 47.1±2.4 | 50.9±1.7 |
| | Medium | 55.2±1.8 | 57.0±0.6 | **60.5±1.1** |
| | Medium-Expert | 41.3±2.9 | 44.2±11.3 | **46.2±10.6** |
| **Walker2D-v2** | Random | 5.3±1.2 | 3.6±1.1 | **13.2±4.9** |
| | Medium-Replay | 90.9±4.9 | 91.2±5.1 | **94.0±4.6** |
| | Medium | 90.5±5.0 | **95.7±2.5** | 88.2±4.7 |
| | Medium-Expert | 114.0±0.8 | **115.3±0.5** | 112.5±0.8 |
| **Hopper-v2** | Random | 9.9±4.4 | 10.8±3.5 | **11.2±2.9** |
| | Medium-Replay | **101.5±1.5** | 95.4±3.3 | 99.7±2.8 |
| | Medium | 100.4±1.0 | **102.2±0.9** | 102.2±3.6 |
| | Medium-Expert | 37.6±33.3 | **39.1±26.4** | 36.5±30.1 |
| **Average** | | 60.4±5.1 | 60.8±5.0 | **62.2±6.0** |

**State.** For behavior policies (AMP and ACL), the state representation includes the current phase, traffic movement efficiency pressure, and the number of effective running vehicles (Zhang et al., 2022). For evaluation algorithms, BCQ and CQL use the same state representation as AMP, while DataLight adopts the number of vehicles, along with the total velocity saturation and unsaturation degrees (Zhang & Deng, 2023).

**Action.** The action is generally defined as the phase selection for the next time period.

**Reward.** AMP, BCQ, CQL and DataLight use pressure (Zhang et al., 2022) as the reward while ACL uses queue length. It is worth noting that we use the opposite of these metrics as the final reward function.

We generate a total of 24K samples for each dataset using the behavior policies for each task. Additionally, we augment 20K samples for each task using our GODA model. The task horizon for each TSC scenario is set to 3600 seconds, with a control step of 15 seconds. For the sampling process, we employ the return-prior goal condition selection method and set the goal scaling factor to 1.2 in TSC tasks. For evaluation methods, we employ the implementation of BCQ, CQL, and DataLight from https://github.com/LiangZhang1996/DataLight.

# C  Further Experimental Results

In this section, we show some more experimental results for our GODA model.

## C.1  Further Evaluation on Gym tasks

To evaluate the generalizability of our proposed method, we further conduct experiments by evaluating GODA with a state-of-the-art offline RL method, i.e., Doubly Mild Generalization (DMG) (Mao et al., 2024), on Gym tasks. DMG introduces a novel generalization strategy by selecting actions from a constrained neighborhood of the training data, thereby maximizing Q-values while avoiding excessive extrapolation. DMG also mitigates generalization propagation and aims to balance generalization and conservatism in offline RL.

As can be seen in Table 9, GODA-enhanced DMG outperforms both the original DMG and the SynthER-enhanced DMG in 7 out of 12 evaluation scenarios, and achieves the highest average normalized score overall. It's worth noting that we were unable to reproduce the results for HalfCheetah-Medium-Expert and Hopper-Medium-Expert for DMG. SynthER and our GODA also fail to improve the results of DMG on these two datasets to normal levels. We suspect this may be due to hyperparameter sensitivity in DMG rather than issues with the datasets themselves. Overall, these results support the claim that GODA can effectively

---

**Algorithm 2** Online Reinforcement Learning with GODA

---

1: **Input:** Real-to-synthetic data ratio $\xi$, diffusion model update frequency $\kappa$
2: Initialize policy $\pi_\phi$, generative model $G_\theta$, real replay buffer $\mathcal{D}^{\text{real}} \leftarrow \emptyset$, synthetic buffer $\mathcal{D}^* \leftarrow \emptyset$
3: **for** $t = 1$ to $T$ **do**
4:     Deploy policy $\pi_\phi$ to collect real transitions and store them in $\mathcal{D}^{\text{real}}$
5:     **if** $t \bmod \kappa = 0$ **then**
6:         Update diffusion model $G_\theta$ using samples from $\mathcal{D}^{\text{real}}$ (see Line 2-5 in Algorithm 1)
7:         Generate synthetic samples using $G_\theta$ and add them to $\mathcal{D}^*$ (see Line 6-11 in Algorithm 1)
8:     **end if**
9:     Update policy $\pi_\phi$ using samples from both $\mathcal{D}^{\text{real}}$ and $\mathcal{D}^*$ with a real-to-synthetic ratio of $\xi$
10: **end for**

---

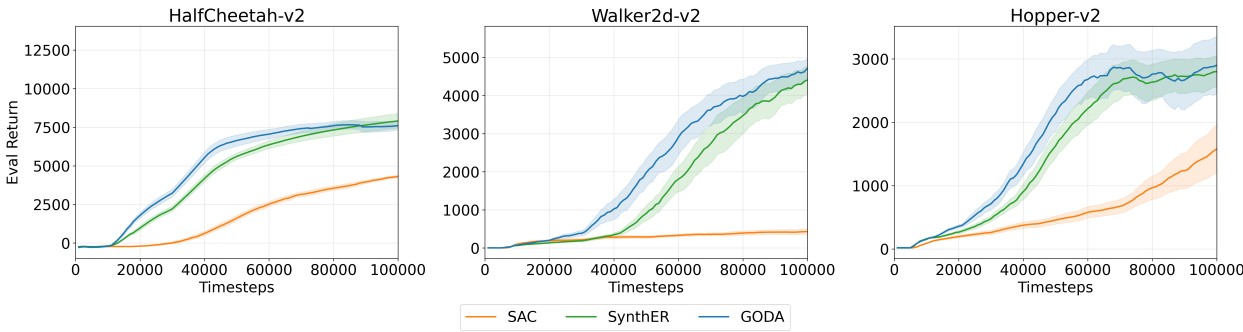

Figure 9: Online evaluation with SAC on Gym tasks. GODA significantly outperforms the baselines in the early stages of training, and continues to match or outperform baselines as training progresses.

generalize to more advanced offline RL algorithms and remains a promising data augmentation framework for enhancing state-of-the-art methods.

## C.2 Online Evaluation

While our primary focus is on data augmentation for offline RL, our framework can be naturally extended to the online RL setting, similar to the approach taken in SynthER (Lu et al., 2024).

Algorithm 2 shows the pseudocode. In this online extension, we periodically train the diffusion model using transitions collected from ongoing environmental interactions. The learned goal-conditioned diffusion model is then used to augment the dataset with synthetic transitions. The online policy is updated using a hybrid buffer containing both real and synthetic transitions in a 1:1 ratio.

To evaluate this setting, we implemented our method using the Soft Actor-Critic (SAC) (Haarnoja et al., 2018) algorithm and conducted experiments on three Gym tasks: HalfCheetah, Hopper, and Walker2D. We train SAC with an Update-To-Data (UTD) ratio of 1 while training SAC (SynthER) and SAC (GODA) with a UTD of 20. For every 10,000 new real data points gathered, we enhance the dataset by augmenting it with 1 million additional synthetic samples.

Because the distribution and quality of the real buffer improve over time during online training, we adapted our hyperparameters accordingly. Specifically, we applied a decaying goal scaling factor, from 1.8 (effective in low-quality, random offline settings) to 1.1 (effective in expert-level offline settings), and an increasing Top-n condition selection, from 50 to 120, to reflect growing confidence in higher-quality goals as training progresses.

We compare SAC (GODA) with both vanilla SAC and SAC (SynthER), and present online learning curves on Figure 9 showing evaluation returns over 100K interaction steps. As shown, GODA significantly outperforms the baselines in the early stages of training, where real transitions tend to be suboptimal. In the later stages, as the quality and quantity of real data improve, GODA continues to match or slightly outperform SAC

(SynthER), with the performance gap gradually narrowing. These results demonstrate that GODA is not only effective in offline settings, but also a promising approach for enhancing online RL through adaptive, goal-conditioned data augmentation.

Table 10: GODA's sensitivity to dataset complexity. We assess the complexity of offline datasets with several Lipschitz constant metrics and use their 95th percentile values to quantify reward sensitivity. The Pen-Human dataset exhibits the highest sensitivity score, making it particularly challenging for GODA to accurately capture the underlying data distribution and effectively extrapolate to higher-reward regions. Bold values indicate the highest sensitivity.

| Dataset | LocalLip95 | GlobalLip95 | GlobalMax | RewardStd | TrajLip95 | GODASensitivity |
|---|---|---|---|---|---|---|
| Door-Cloned | 3.361 | 0.016 | 5.9 | 4.0313 | 7.447 | 10.16 |
| Door-Human | 3.016 | 1.798 | 4.8 | 5.6160 | 4.757 | 13.45 |
| Pen-Cloned | **29.728** | **15.896** | 796.5 | 28.3944 | 325.811 | 106.03 |
| Pen-Human | 20.626 | 9.450 | **1474.1** | **29.0234** | **407.446** | **109.88** |

## C.3 Failure Case Analysis of GODA

We observe that GODA fails to perform well on some scenarios, such as the Adroit Pen-Human dataset, as shown in Table 3. Therefore, we develop a comprehensive reward sensitivity measurement to examine which offline RL dataset characteristics make GODA effective or problematic. Our method quantifies reward landscape sensitivity using multiple Lipschitz constant metrics across four D4RL Adroit datasets. *Local Lipschitz constants* are computed using k-nearest neighbors and weighted least squares to estimate how rapidly rewards change with small state-action perturbations at each data point. *Global Lipschitz analysis* calculate pairwise ratios of reward differences to distances across all neighbor pairs to capture worst-case sensitivity spikes throughout the dataset. *Trajectory-level analysis* examine within-trajectory sensitivity by computing maximum Lipschitz ratios between trajectory steps. These metrics were combined into a composite *GODA Sensitivity* weighted by local sensitivity (40%), global sensitivity (30%), reward variability (20%), and trajectory sensitivity (10%), with data normalization and efficient subsampling for computational tractability.

$$L_r = \sup_{(s,a) \neq (s',a')} \frac{|r(s,a) - r(s',a')|}{||(s,a) - (s',a')||_2} \tag{13}$$

As can be seen in Table 10, the analysis revealed that task complexity plays an important role on GODA's success. Door manipulation tasks achieved excellent GODA Sensitivity with low local Lipschitz constants, stable global behavior, and manageable trajectory sensitivity, making them highly preferable for GODA. Conversely, pen manipulation tasks exhibited catastrophic sensitivity with GODA Sensitivity exceeding 100, extreme local Lipschitz values, global sensitivity spikes reaching 800-1500, and volatile trajectory patterns, making them fundamentally challenging for GODA. The critical insight is that the difference between Human and Cloned datasets are minimal within the same task type, but task-intrinsic reward sensitivity creates a significant gap. GODA's failure on Pen-Human may also stem from the fact that the original dataset contains only 25 human demonstration trajectories, which lack sufficient diversity for GODA to effectively learn the underlying data distribution and extrapolate to out-of-distribution high-reward regions.

## C.4 Ablation on Scaling Factor on Different Datasets

To evaluate the practical effectiveness of goal scaling, we conducted additional ablation studies on the Walker2D-Random, Medium, and Medium-Expert datasets.

Figure 10 shows that goal scaling does have a tangible impact: On the Random dataset, performance improved significantly with scaling factors of 1.8 for IQL and 1.4 for TD3+BC. For instance, the score gap between $\lambda = 1.8$ and $\lambda = 1.0$ in IQL reached 14.6. On Medium and Medium-Replay, a scaling factor of 1.1 consistently yielded the best results for IQL, with gains up to 4.2 compared to $\lambda = 1.0$. In contrast, for the Medium-Expert dataset, the difference between $\lambda = 1.0$ and $\lambda = 1.1$ was marginal. This is likely because datasets with more expert demonstrations have less room for improvement, and higher scaling can push goals outside the feasible return range.

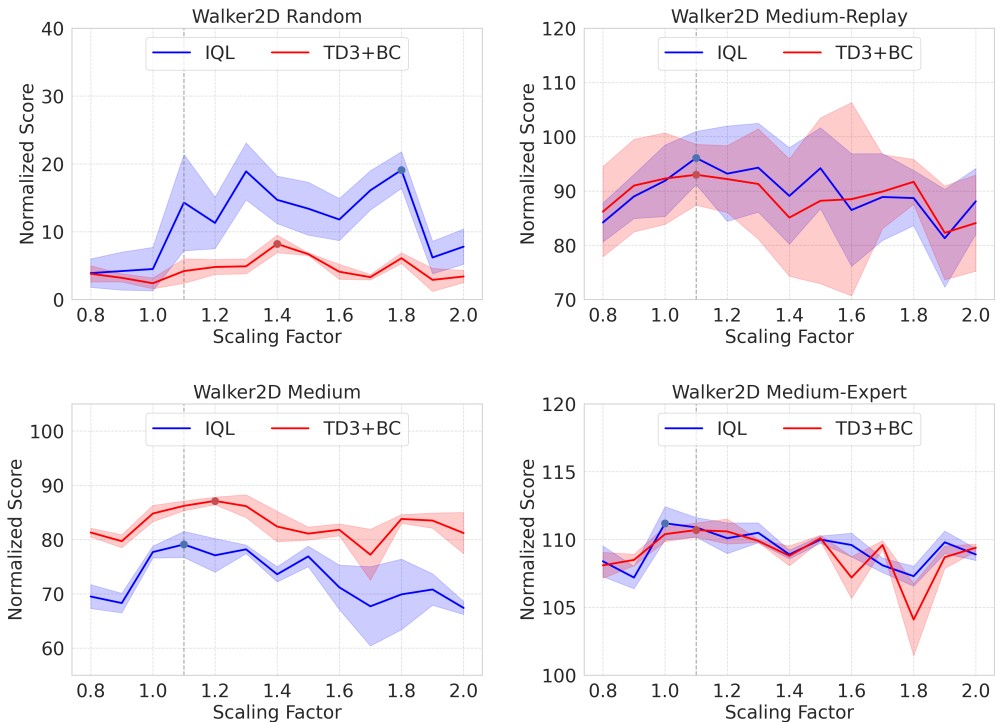

Figure 10: Ablation on scaling factor on different datasets. Goal scaling is particularly beneficial in lower-quality datasets while still necessary for higher-quality datasets.

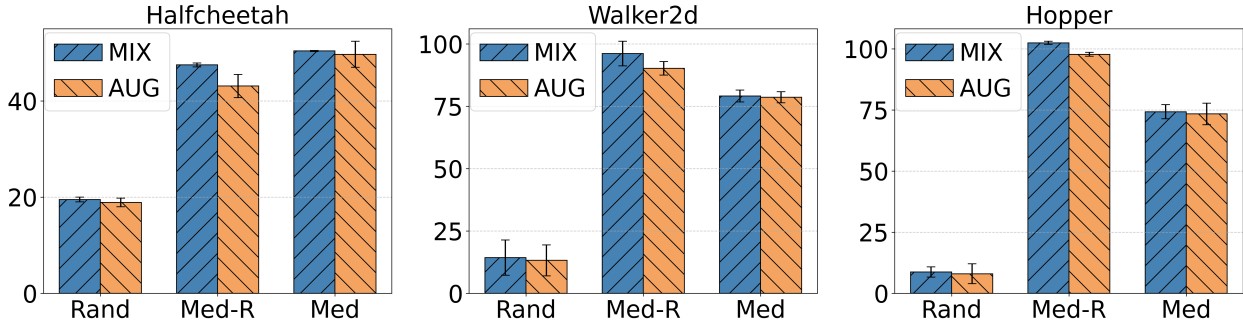

Figure 11: Ablation on mixed datasets for IQL evaluation.

These results suggest that goal scaling is particularly beneficial in low-quality datasets, where it enables exploration of high-return regions beyond what the original data distribution offers. While $\lambda = 1.1$ performs robustly across all quality levels and is used as our default, further gains can be achieved through task-specific hyperparameter tuning.

## C.5 Ablation on Mixed Datasets

Since GODA primarily augments samples from high-reward regions of the data distribution, which might result in a lack of diversity, we use a mix of both the original and augmented datasets for training. In this section, we compare the performance of our default setting (mixed datasets) with the use of only augmented datasets. As shown in Figures 11, removing the original datasets leads to slight performance degradation across most tasks. Therefore, combining our augmented datasets with the original datasets helps increase the diversity and extend the reward boundary.

Table 11: Ablation study on mixed datasets for baseline methods.

| Task | Dataset | IQL | | | | TD3+BC | | | |
|------|---------|-----|-----|-----|------|--------|-----|-----|------|
| | | TATU | SynthER | DStitch | GODA | TATU | SynthER | DStitch | GODA |
| **Gym Avg.** | w/o Original Data | 46.2±4.3 | 52.1±2.4 | 51.5±1.5 | **52.5±2.9** | 43.3±4.2 | 46.3±4.7 | **47.9±3.3** | 47.0±4.6 |
| | w/ Original Data | 46.0±3.5 | 51.2±4.4 | 52.1±1.6 | **54.7±2.2** | 41.8±7.9 | 46.2±4.5 | 46.8±4.6 | **48.4±3.9** |
| **Maze2D Avg.** | w/o Original Data | 45.7±8.2 | 45.6±2.9 | 47.5±6.8 | **66.7±9.3** | 68.2±10.6 | 65.1±29.7 | 65.9±24.9 | **74.5±11.2** |
| | w/ Original Data | 47.1±6.8 | 44.8±9.2 | 46.0±8.4 | **68.1±6.6** | 68.0±10.9 | 65.5±37.6 | 65.1±23.7 | **79.1±7.5** |

Figure 12: Performance comparison on JN1 task with difference sizes of original real-world training datasets. The parts exceeding the maximum travel time are not displayed.

We further present the results for baseline data augmentation methods when combining original datasets with augmented datasets, as shown in Table 11. For Gym tasks, TATU and SynthER exhibit degraded performance when the original datasets are included during training, while their performance remains comparable on Maze2D tasks. In contrast, DStitch demonstrates the opposite trend, performing comparably on Gym tasks but worse on Maze2D tasks when using the mixed datasets. Our GODA shows reduced scores when the original datasets are removed; however, its final results remain superior to other baselines. GODA only delivers slightly worse performance than DStitch on Gym tasks when evaluated with TD3+BC, but still outperforms TATU and SynthER across all tasks.

## C.6 Ablation on Size of Original Dataset

Our experiments on TSC tasks utilize only 24K samples, compared to 1–2M samples for D4RL tasks. The superior performance across both task types demonstrates that GODA not only excels in tasks with large training datasets but also addresses real-world challenges effectively with much smaller datasets. Considering the even greater limitations of original datasets in real-world TSC scenarios, we further evaluate GODA on TSC datasets with reduced sizes, ranging from 24K to 2.4K samples.

Figure 12 illustrates the performance of different evaluation algorithms trained separately on the original datasets and on the mixture of original and augmented datasets. Notably, we only reduce the size of the original datasets used for directly training the evaluation algorithms or for training GODA, while maintaining the same number of augmented samples (20K) across all cases. Specifically, for the "Original" method, the evaluation algorithms are trained solely on the reduced original datasets. For GODA, the GODA model is first trained on the reduced original datasets, and then 20K augmented samples are generated. The evaluation algorithms are subsequently trained on the mixed datasets.

The results indicate that when the original training datasets contain fewer than 9.6K samples, all three evaluation algorithms trained directly on the reduced original datasets fail to learn effective policies, though their performance improves as the dataset size increases. In contrast, GODA effectively augments high-quality data, enabling the training of qualified evaluation policies for most sizes of original datasets. Remarkably, GODA achieves this even with only 2.4K samples in the original datasets, failing in just one case each for BCQ and CQL, while consistently demonstrating strong performance otherwise.

