# OpenReview forum: "Goal-Conditioned Data Augmentation for Offline Reinforcement Learning"
_TMLR — Accepted by TMLR_

### Review · Reviewer_YYPb · 2025-06-07

**Summary Of Contributions:**

The paper proposes a diffusion-based data augmentation algorithm for offline RL called Goal-cOnditioned Data Augmentation (GODA). GODA trains a diffusion model conditioned on return-to-go (RTG) and employs a novel goal-selection mechanism and controllable scaling to guide the model towards producing near-optimal samples with high RTGs. The paper also introduces new neural network architectures, including the adaptive gated long skip connection and the gated residual MLP block. GODA outperforms existing data augmentation offline RL approaches on various benchmarks. Extensive ablation studies were conducted to demonstrate that every component of the algorithm contributes to the overall performance.

**Audience:**

Yes

**Broader Impact Concerns:**

I do not have any concerns about the ethical implications of the work.

**Claims And Evidence:**

Yes

**Requested Changes:**

All of the following are non-critical.

1. The paper is rather long. Please consider moving some unnecessary parts, such as hyperparameter settings, implementation details, and environment details, to the appendix.

2. The last sentence, "It is worth noting that a ... rewards unchanged[,]" on page 6 is difficult to understand.

3. In computing the L2 distance metric (11), it is unclear how the two samples $(s_i, a_i)$ and $(\bar{s}_i, \bar{a}_i)$ are selected. Lu et al. (2024) appear to compute the minimum distance, but the paper does not explicitly state this.

4. The authors have used textual citations for all of their citations, but most of them should be parenthetical instead. For example, in the first sentence of the Introduction, "Reinforcement Learning Sutton & Barto (2018)" should be changed to "Reinforcement Learning (Sutton & Barto, 2018)".

5. In the second last sentence on page 7, "... introduce a condition-adaptive gated residual connection to further enhances the ..." should be changed to "... introduce a condition-adaptive gated residual connection to further [enhance] the ..."

**Strengths And Weaknesses:**

The proposed algorithm, GODA, is a simple and effective method for generating high-quality synthetic data for offline RL. The authors conducted experiments on multiple benchmarks, and GODA has outperformed existing data-augmentation baselines on most of them. They also performed extensive ablation studies, which helped the reader understand the contribution of each component. However, there is room for improvement in terms of clarity and formatting, which I will discuss in detail in the next section. Also, it would be interesting to see the contribution of adaptive gated long skip connection, as the paper only provides a comparison between GODA, adaLN, and in-context conditioning.

---

> ### Author Response · Authors · 2025-06-23
> **Responses to Comments: Part 1**
>
> Thanks for your valuable time and insightful comment! These comments are quite helpful.
>
> Following your comments, we have conducted further ablation experiments on both the condition-adaptive gated long skip connection (Section 4.4.2) and condition-adaptive gated residual connection (Section 4.4.3) and each adaptive gating component as well.
>
> We present the results in the following table, where **AdaLN** removes both the *adaptive gated long skip connection* and *adaptive gated residual connection* but only retains the AdaLN structure; **w/o AG-LSC** removes the *adaptive gated long skip connection*; **w/o AG-RC** removes the *adaptive gated residual connection*; **w/o Gate-LSC** retains the long skip connection, but without any *gating* mechanism; and **w/o Gate-RC** retains the residual connection, but without *gating*.
>
> **Table: Ablation study on adaptive gated long skip connection and adaptive gated residual connection.**
>
> | **Task**    | **Eval Alg.** | **AdaLN** | **w/o AG-LSC** | **w/o AG-RC** | **w/o Gate-LSC** | **w/o Gate-RC** | **GODA**     |
> | ----------- | ------------- | --------- | -------------- | ------------- | ---------------- | --------------- | ------------ |
> | **WK2D**    | IQL           | 89.1±5.7  | 90.2±8.2       | 94.2±6.1      | 92.4±8.6         | 95.0±11.7       | **96.1±9.4** |
> |             | TD3+BC        | 91.7±5.9  | 92.2±4.9       | 92.5±7.9      | 91.9±6.5         | 92.9±9.0        | **93.0±5.6** |
> | **Average** | —             | 90.4±5.8  | 91.2±6.6       | 93.4±4.0      | 92.2±7.6         | 94.0±10.4       | **94.6±7.5** |
>
> These results show that removing either the adaptive gated long skip connection or the adaptive gated residual connection leads to performance degradation, with the adaptive gated long skip connection presenting a more pronounced impact. Replacing the adaptive gated long skip connection with a vanilla long skip connection, i.e., without any adaptive gating, also results in a significant performance drop, though it still performs slightly better than having no long skip connection at all. This highlights the importance of incorporating goal information into the conditioning structure. A similar trend is also observed for the adaptive gated residual connection.
>
> **Responses for your additional comments:**
>
> 1. Thanks for your valuable comments! We will move some parts, including hyperparameter settings, implementation details, some environmental settings, and some less important experimental results to the appendix.
>
> 2. Thanks for pointing this out. The “reward at first timestep” should be changed to “reward at the last timestep.” We will provide a more detailed explanation as follows:
>
> "It is worth noting that we use a goal scaling factor rather than an increment factor, i.e., `goal = (ĝₜ + λ, t)`. The reason is that incrementing each RTG `ĝₜ = ∑_{t'=t}^{T} r_{t'}` with a fixed value would be equivalent to simply incrementing the terminal reward `r_T` by `λ` while leaving all other target rewards `r_t` (for `t ≠ T`) unchanged.
> However, assigning a higher RTG goal at each timestep is intended to encourage the agent to generate trajectories that yield higher rewards throughout the entire trajectory, rather than only improving the terminal reward. Therefore, a **goal scaling factor** can be a better choice than an increment factor."

---

> ### Author Response · Authors · 2025-06-23
> **Responses to Comments: Part 2**
>
> Continuing from the previous Part 1
>
> 3. Thank you very much for your insightful comment. We acknowledge that we originally missed the summation symbol in Equation (11) for the L2 distance metric. We have updated the equation in our revised version as follows:
>
> $\text{Min L2 Distance} = \frac{1}{M} \sum_{i=1}^{M} \left\| (s_{t}^{i}, a_{t}^{i}) - (\bar{s}_{t}^{i}, \bar{a}_{t}^{i}) \right\|_2$
>
> Additionally, we did not use the minimum L2 distance approach as adopted in the SynthER paper. Instead, we randomly sampled 10,000 transitions from the original dataset, SynthER-augmented, and GODA-augmented datasets, and computed the average Euclidean distance between the augmented samples and the original dataset samples without identifying the nearest neighbors. However, we do agree that adopting the minimum L2 distance strategy from SynthER, i.e., calculating the average L2 distance between each augmented sample and its nearest neighbor in the original dataset, is a more appropriate and informative metric for evaluating data quality. We have thus recomputed the results using this metric and updated them in the revised version of the paper. Updated figures will be presented in our revised version.
>
> **Table: Data quality evaluation metrics for SynthER and GODA on Walker2D tasks.**
> *Smaller Dynamics MSE, larger Min L2 Distance, and larger Average Reward indicate better quality.*
>
> | **Task**                  | **Dynamics MSE (SynthER)** | **Dynamics MSE (GODA)** | **Min L2 Distance (SynthER)** | **Min L2 Distance (GODA)** | **Average Reward (SynthER)** | **Average Reward (GODA)** |
> | ------------------------- | -------------------------- | ----------------------- | ----------------------------- | -------------------------- | ---------------------------- | ------------------------- |
> | Walker2D-Random-v2        | 2.7±5.7                    | **1.9±2.9**             | 1.9±0.6                       | **2.4±0.6**                | 0.1±0.6                      | **0.6±0.5**               |
> | Walker2D-Medium-Replay-v2 | 0.5±1.7                    | **0.4±1.1**             | 1.6±0.9                       | **1.7±0.7**                | 2.5±1.3                      | **3.5±0.9**               |
> | Walker2D-Medium-v2        | **0.3±1.0**                | **0.3±0.8**             | **0.8±0.5**                   | **0.8±0.6**                | 3.4±1.2                      | **3.7±0.9**               |
>
> These new results show that the Min L2 Distance follows the same trend as the original L2 Distance metric. Specifically, GODA exhibits significantly lower Dynamics MSE and a broader range of Min L2 Distance values, indicating better alignment with the underlying environmental dynamics and higher sample diversity. Notably, GODA is particularly effective on random-level datasets with high randomness and sparse optimal demonstrations, where traditional augmentation methods often fall short.
>
> 4. Thanks for pointing this out. We have further proofread the manuscript and fixed the citation format for all the citations in our revised version.
>
> 5. Thank you for pointing this out. We have corrected the grammatical error in our revised version.

---

> ### Author Response · Authors · 2025-06-23
>
> We greatly appreciate your insightful comments and suggestions. We hope our responses and revisions have resolved your concerns and improved the clarity of the manuscript.

---

### Review · Reviewer_ZSer · 2025-06-29

**Summary Of Contributions:**

The performance of offline reinforcement learning (RL) methods is dependent on the quality of the collected data (in contrast to online RL), which in most cases can lead to suboptimal policies. This paper introduces GODA, an approach that augment the data using the output of a conditional diffusion model that model the distribution of the transitions given some metric to generate high-return data. The authors did a good job in demonstrating the higher data quality generated by GODA and the better performance on various tasks.

**Audience:**

Yes

**Claims And Evidence:**

Yes

**Requested Changes:**

- The paper lacks a **discussion on the limitations** of their approach. I think it's a critical to the paper and I request that the authors to add it. For example, after reading the paper, it's not clear to me how general your approach can be. Can it be applied to visual RL datasets or robot datasets? If not, what can possibly be improved or modified to support those complex tasks? etc.

- **Improve clarity of writing of the methods section** to make it easier to follow. This is a critical issue that needs to be addressed.

- I think the paper is generally easy to follow although it's rather long without a good reason for the length. I suggest the authors try to make writing more concise and reduce the size of the manuscript. This is only a recommendation and not a critical issue.

**Strengths And Weaknesses:**

The paper addresses an important problem in offline reinforcement learning which is the learned policy is limited to the quality of the data. The authors introduces GODA, an approach that generate new high-return data to improve the effectiveness of the learned policies. This is relevant for many applications and has a wide audience. I think the paper is generally easy to follow (except the method's section which needs a lot of work to improve its clarity).

The main issue in the paper is the absence of the limitations section.

---

> ### Author Response · Authors · 2025-07-14
>
> •	*The paper lacks a discussion on the limitations of their approach. Can it be applied to visual RL datasets or robot datasets?*
>
> **Response:**
>
> Thank you for raising this important point. For *robot datasets*, we have conducted experiments on Adroit tasks, which involve manipulating a pen and opening a door using a 24-DoF simulated hand robot. The results were presented in Table 5 and Section 7.1.3 of the first version.
>
> We acknowledge that our current method is not directly applicable to high-dimensional *pixel-based visual RL datasets*. This is because our diffusion-based data augmentation operates in a low-dimensional feature space, whereas generating large volumes of raw image data directly with a diffusion model is both computationally expensive and technically challenging. However, to extend our approach to such domains, one possible solution is to follow the strategy adopted in SynthER [1]. In their approach, the original dataset is first used to pre-train a Behavior Cloning (BC) policy network composed of a CNN encoder, a trunk network, and a final fully connected layer. The CNN encoder and trunk network are then frozen and used to transform each observation and next observation, represented as raw image data of size 84×84×3, into a 50-dimensional latent representation. These low-dimensional representation data, combined with rewards and actions, are then used to train the diffusion model and augment transition samples in the latent space. These augmented samples are subsequently used to fine-tune only the final fully connected layer (output head), thereby avoiding the need to directly generate high-dimensional image data.
>
> We follow the same settings and augment 5M latent transitions for each data quality level to evaluate our GODA on the Cheetah-Run task from the pixel-based V-D4RL [2] benchmark. The following table shows the results of BC algorithm evaluated on pixel-based datasets.
>
> **Table**: Evaluation of GODA on the Cheetah-Run task from the pixel-based V-D4RL benchmark using the BC algorithm, with sample augmentation performed in the latent space.
>
> | **Eval Algorithm** | Aug Methods | **Rand**   | **Med-R**     | **Med**      | **Med-E**      | **Average**   |
> |--------------------|--------------|------------|----------------|---------------|----------------|----------------|
> | BC                 | Original     | 0.0±0.0    | 25.0±3.6       | 51.6±1.4       | 57.5±6.3        | 33.5±2.8       |
> |                    | SynthER      | 0.1±0.0    | 27.9±3.4       | 52.2±1.2       | **69.9±9.5**    | 37.5±3.5       |
> |                    | **GODA**     | **1.4±0.8**| **35.5±6.1**   | **54.9±2.0**   | 67.6±10.4       | **39.9±4.8**   |
>
> The results indicate that GODA achieves the best performance on three out of four dataset quality levels and obtains the highest average score. It outperforms the BC algorithm without data augmentation by 19.1%, and the BC with SynthER-augmented samples in the latent space by 6.4%. Notably, GODA delivers substantial improvements on the Random and Medium-Replay datasets, highlighting its suitability for offline scenarios with limited expert demonstrations.
>
> However, we also agree that a discussion on the *limitations* and generality of our approach is essential. We have now added a subsection in the Discussion section of the revised manuscript that explicitly addresses the limitations.
>
> *We acknowledge that our current method is not directly applicable to high-dimensional, pixel-based visual RL or real-world robotics datasets, as it requires transforming image data into low-dimensional latent representations before applying diffusion-based data augmentation. Since our approach operates in a low-dimensional feature space, this additional transformation step can complicate the implementation and introduce dependencies on the architecture of the policy network. In future work, we aim to develop more direct methods for augmenting pixel-based datasets by improving the sampling efficiency of the diffusion model. We also plan to explore the application of GODA to more complex, real-world scenarios.*
>
> [1] Lu, Cong, et al. "Synthetic experience replay." Advances in Neural Information Processing Systems 36 (2023): 46323-46344.
>
> [2] Lu, Cong, et al. "Challenges and opportunities in offline reinforcement learning from visual observations." Transactions on Machine Learning Research (2023).

---

> ### Author Response · Authors · 2025-07-14
>
> •	*Improve clarity of writing of the methods section to make it easier to follow. This is a critical issue that needs to be addressed.*
>
> **Response:**
>
> Thank you for your valuable feedback. We have carefully revised the Methods section to enhance clarity and improve readability. Specifically, we made the following changes:
>
> 1.	**New Illustrative Figure for Method Overview:** We have added a figure in Section 3 to provide a clear overview of the proposed method, illustrating both the training and data augmentation processes.
>
> 2.	**High-Level Overview:** We revised the introductory paragraph of the Methods section to provide a clearer high-level overview of our approach. It now includes brief introductions to each component and references to the corresponding subsections, helping to guide the reader through the structure of our method.
>
> 3.	**Variable Definitions:** We added explicit descriptions and signals for all variables introduced in the methodology to minimize potential confusion.
>
> 4.	**Algorithm 1 Improvements:** We refined the pseudocode in Algorithm 1, added detailed comments, and reorganized its structure to improve clarity and facilitate understanding.
>
> 5.	**Expanded Explanation of Algorithm 1:** In Section 4.5, we added more detailed explanations of Algorithm 1, making the processes of diffusion model training, data augmentation, and policy training and evaluation easier to follow.
>
> 6.	**New Illustrative Diagrams:** In Section 4.2, we added two illustrative diagrams to clarify how goal conditions are selected under different paradigms.
>
> 7.	**Figure of denoiser neural network Refinement:** We revised Figure 2 (The denoiser neural network) and its caption to enhance clarity and improve the expressiveness of the visual representation.
>
> We hope these revisions address your concern and improve the overall readability of the Methods section. All of the revisions will be highlighted in blue in our revised version.
>
> •	*I think the paper is generally easy to follow although it's rather long without a good reason for the length. I suggest the authors try to make the writing more concise and reduce the size of the manuscript. This is only a recommendation and not a critical issue.*
>
> **Response:**
>
> Thank you very much for the suggestion.  In the revised manuscript, we have made our best efforts to reduce the length of the manuscript. Specifically, we removed repetitive statements, streamlined the writing for conciseness, and relocated several less critical components to the Appendix, including the Related Work section, the Hyperparameters subsection, certain experimental details, and some ablation study results.
>
> We sincerely appreciate your thoughtful feedback. We hope our revisions have addressed your concerns and improved the manuscript’s clarity.

---

### Review · Reviewer_zNDP · 2025-07-13

**Summary Of Contributions:**

The authors  propose a goal-conditioned data augmentation (GODA) framework, which augments offline reinforcement learning (RL) datasets with synthetically generated samples of higher quality. The generative model is based on the Elucidated diffusion model (EDM), which additionally conditions on a return-oriented goal. To account for noisy inputs and conditions GODA relies on an adaptive gated conditioning method, which significantly enhances the ability to guide the diffusion and sampling process. The idea is for GODA to learn the distribution of the original offline dataset, while generating new data of higher quality/return. This can improve the performance of offline RL, especially in the presence of small and suboptimal offline datasets. Experiments on the D4RL benchmark and traffic signal control tasks demonstrate the superior performance of the proposed framework compared to the state-of-the-art data augmentation methods.

**Audience:**

Yes

**Broader Impact Concerns:**

I do not have any broader concerns that are not addressed in the manuscript.

**Claims And Evidence:**

Yes

**Requested Changes:**

I generally do not have major concerns with this work. However, I would appreciate it if the authors could respond to the previous questions, as well as to some new ones:
- In Table 4, HP with Med-R has extremely different performance under IQL than under TD3+BC. Why is there such a dramatic gap between the two algorithms? Fora RAND setting, I would have understood such a fluctuation, but what about the Med-R setting? Does that perhaps show that Med-R is inherently unstable?
- I was wondering whether this framework could be used even in the context of online RL to get higher-quality data. The main difference would be that Algorithm 1 would be run periodically, and we would periodically re-train the goal-conditioned neural net. Have the authors considered such a use case?
- IQL and TD3+BC are not very recent methods. Have the authors considered more recent baselines? It would be interesting to compare the different settings under state-of-the-art offline RL methods.
- It would have been interesting to see experiments with High-quality datasets, as opposed to Med and Med-R. This would allow one to see whether goal-conditioned data augmentation can raise Med or Med-R performance close to the optimum, or whether the gap remains. Ideally, we would like to reach a performance similar to a high-quality offline dataset after goal-conditioned data augmentation. Is that really what is happening? Insights into that could be very helpful.

**Strengths And Weaknesses:**

Strengths
- The idea of goal-conditioned data generation is meaningful and well-motivated in the RL setting, where high quality transitions could in principle make a difference.
- EDM is a solid choice for the base diffusion model. The proposed optimizations, such as goal-conditioning, adaptive gated conditioning, and selective goal conditions are interesting and have some degree of novelty.
- The experiments contained both standard tasks as well as tasks inspired by the real world (TSC). The evaluation shows quite clearly and convincingly that the proposed methods usually outperform other data augmentation methods, under both IQL and TD3+BC. The figures also show how GODA is able to sample from the high-return regions, promoting higher quality but also enhanced diversity.
- GOAD can perform well even under small datasets, as results on TSC demonstrate.
- The ablation study is extensive and provides insights into various practical aspects of the proposed framework and algorithms.

Weaknesses
- There is no theoretical guarantee of any sort. This is to be expected to some extent, given the inherent difficulty and uncertainty of offline RL. That being said, there are settings where GODA performs worse than the state of the art (e.g., on Pen-Androit). It would be great, if the authors could perhaps provide possible insights into the settings that are more challenging to GODA. Perhaps settings where small perturbations in the transition can result in very different rewards, or something along these lines?
- I was not so clear about the controllable goal scaling part. In practice, it seems values close to 1 are good enough. Does it even make a difference if we just always set it to 1? I mean, intuitively Equation (6) makes sense, but in practice I got the impression that a value of 1 will typically be close to optimal, so that scaling by some $\lambda$ is not really necessary. Similarly, I was not sure whether the Top n conditions make a big difference, especially given the high variance that we observe in Figure 7.
- I think it would have been nice to get more explanations/insights into some of the experimental results. For instance: (a) Why the return-prior generally outperforms the RTG-prior? The latter intuitively seems a strong baseline, so what could be the possible reasons for that? (b) Why does DStitch suffer from more diverse datasets? (c) Why does sometimes Med-R perform better than Med, and other times the opposite trend  is observed? For instance, Med-R is worse than Med for HC but better for WK2D (with IQL, Table 4). Do the authors have insights into which type of dataset is preferable and why?

---

> ### Author Response · Authors · 2025-08-01
>
> We sincerely thank you for your insightful comments. We have addressed them and highlighted all revisions in blue in the updated manuscript.
>
> (1) *There is no theoretical guarantee of any sort. This is to be expected to some extent, given the inherent difficulty and uncertainty of offline RL. That being said, there are settings where GODA performs worse than the state of the art (e.g., on Pen-Androit). It would be great, if the authors could perhaps provide possible insights into the settings that are more challenging to GODA.*
>
> **Response**
>
> Since our work focuses on data augmentation rather than altering the underlying offline RL algorithms, the theoretical foundation of our method lies primarily in the conditional diffusion model we employ. Specifically, we build upon the Elucidated Diffusion Model (EDM), which offers well-established theoretical support through denoising score matching and ODE-based sampling (see Section 2.2). Our goal-conditioned extension enhances this foundation by introducing an adaptive gated conditioning mechanism, which integrates return-to-go and timestep signals into the denoising process. Additionally, our controllable goal scaling strategy provides a principled means of guiding the sampling process toward higher-reward regions.  Furthermore, Figures 1 and 7 also present the motivations and showcase the intuitions of our work. We also empirically validate the quality and stability of our augmented data using metrics such as Dynamics MSE, min L2 Distance and Average Reward.
>
> However, we totally agree that additional analysis on what settings are preferable or challenging for our GODA is important. We further examine GODA’s underperformance in challenging settings like Pen-Human. Specifically, Adroit-Pen task involves high-dimensional, dexterous manipulation, where small perturbations in action can lead to disproportionately large changes in reward (e.g., minor deviations in gripper position can affect pen stability). We use statistical analysis to support this in **Section C.3 in Appendix**.
>
> Concretely, we used a comprehensive reward sensitivity measurement to evaluate which offline RL dataset characteristics make GODA effective or problematic. Our method quantifies reward landscape sensitivity using multiple Lipschitz constant metrics across four D4RL Adroit datasets (pen-human, pen-cloned, door-human, door-cloned) and use their 95th percentile values to quantify reward sensitivity. **Local Lipschitz constants** were computed using k-nearest neighbors and weighted least squares to estimate how rapidly rewards change with small state-action perturbations at each data point. **Global Lipschitz analysis** calculated pairwise ratios of reward differences to distances across all neighbor pairs to capture worst-case sensitivity spikes throughout the dataset. **Trajectory-level analysis** examined within-trajectory sensitivity by computing maximum Lipschitz ratios between trajectory steps. These metrics were combined into a composite **GODA Sensitivity** weighted by local sensitivity (40%), global sensitivity (30%), reward variability (20%), and trajectory sensitivity (10%), with data normalization and efficient subsampling for computational tractability.
>
> **Table**: GODA's sensitivity to dataset complexity. Bold values indicate the highest sensitivity.
>
> | **Dataset**      | **LocalLip95** | **GlobalLip95** | **GlobalMax** | **RewardStd** | **TrajLip95** | **GODASensitivity** |
> |-|-|-|-|-|-|-|
> | **Door-Cloned** | 3.361  | 0.016  | 5.9   | 4.0313 | 7.447 | 10.16 |
> | **Door-Human**       | 3.016  | 1.798  | 4.8 | 5.6160| 4.757| 13.45 |
> | **Pen-Cloned** | **29.728**| **15.896** | 796.5| 28.3944 | 325.811| 106.03|
> | **Pen-Human**  | 20.626| 9.450| **1474.1**| **29.0234**| **407.446** | **109.88** |
>
> The analysis revealed that task complexity plays a critical role on GODA’s success. Door manipulation tasks achieved excellent GODA Sensitivity with low local Lipschitz constants, stable global behavior, and manageable trajectory sensitivity, making them highly preferable for GODA. Conversely, pen manipulation tasks exhibited catastrophic sensitivity with GODA Sensitivity exceeding 100, extreme local Lipschitz values, global sensitivity spikes reaching 800-1500, and volatile trajectory patterns, making them fundamentally challenging for GODA. The critical insight is that the difference between Human and Cloned datasets are minimal within the same task type, but task-intrinsic reward sensitivity creates a significant gap. GODA’s failure on Pen-Human may also stem from the fact that the original dataset contains only 25 human demonstration trajectories, which lack sufficient diversity for GODA to effectively learn the underlying data distribution and extrapolate to out-of-distribution high-reward regions.
>
> We appreciate you raising this point. We have incorporated these into Section 6.1.1, 7, and C.3 in the revised manuscript.

---

> ### Author Response · Authors · 2025-08-01
>
> (2) *I was not so clear about the controllable goal scaling part. In practice, it seems values close to 1 are good enough. Does it even make a difference if we just always set it to 1? I mean, intuitively Equation (6) makes sense, but in practice I got the impression that a value of 1 will typically be close to optimal, so that scaling by some λ is not really necessary. Similarly, I was not sure whether the Top n conditions make a big difference, especially given the high variance that we observe in Figure 7.*
>
> **Response**
>
> Thank you for raising these insightful points about the practical relevance of the controllable goal scaling parameter (λ) and the impact of Top-$n$ conditions.
>
> **On the goal scaling parameter (λ)**:
>
> To evaluate the practical effectiveness of goal scaling, we have conducted additional ablation studies on the Walker2D-Random, Medium, and Medium-Expert datasets (included in **Figure 10 in the revised Appendix**). Results for the Medium-Replay dataset are already provided in Figure 5 of the main text. Our findings showcase that goal scaling does have a tangible impact:
>
> •	On the Random dataset, performance improved significantly with scaling factors of 1.8 for IQL and 1.4 for TD3+BC when compared to λ=1.0. Specifically, the normalized score improves by 14.6 when λ=1.8 compared with λ=1.0 in IQL evaluation. In TD3+BC, the score rises from 2.4 (λ = 1.0) to 8.2 (λ = 1.4).
>
> •	On Medium and Medium-Replay, a scaling factor of 1.1 consistently yielded the best results for IQL, with gains up to 4.2 compared to λ=1.0.
>
> •	In contrast, for the Medium-Expert dataset, the difference between λ=1.0 and λ=1.1 was marginal. This is likely because datasets with more expert demonstrations have less room for improvement, and higher scaling can push goals outside the feasible return range.
>
> These results suggest that goal scaling is particularly beneficial in low-quality datasets, where it enables exploration of high-return regions beyond what the original data distribution offers. While λ=1.1 performs robustly across all quality levels and is used as our default, further gains can be achieved through task-specific hyperparameter tuning.
>
> **On the Top-$n$ goal selection**:
>
> While this hyperparameter may seem less critical than λ, it can still meaningfully influence performance. For example, as shown in Figure 7, the best and worst Top-$n$ settings on Walker2D Medium-Replay using IQL yield scores of 96.1 and 87.3, respectively, resulting in a notable difference of 8.8. The Top-$n$ selection primarily acts as a filter, ensuring that data augmentation focuses on high-return regions. However, increasing $n$ can introduce more diversity, allowing the diffusion model to learn from a broader range of transitions. This represents a classic trade-off between data quality and diversity: smaller $n$ prioritizes high-return samples, while larger $n$ can improve generalization through diversity. This is conceptually similar to the contrast between D4RL’s Medium (high-quality, low-diversity) and Medium-Replay (lower-quality, higher-diversity) datasets, where different algorithms excel depending on their strengths.
>
> **Regarding the high variance observed**:
>
> For Walker2D-Random, the high variance is likely due to the inherent instability and noisiness of the original dataset, compounded by the stochastic nature of the diffusion model used for augmentation.
>
> For Walker2D-Medium-Replay, the variance appears to stem more from the dataset itself and the behavior of the evaluation algorithms, as we observe high variance across all methods evaluated on this benchmark.
>
> We have incorporated these clarifications into the revised version to better explain the roles and effects of both the goal scaling and Top-$n$ parameters.

---

> ### Author Response · Authors · 2025-08-01
>
> (3) *I think it would have been nice to get more explanations/insights into some of the experimental results. For instance: (a) Why the return-prior generally outperforms the RTG-prior? The latter intuitively seems a strong baseline, so what could be the possible reasons for that?*
>
> **Response**
>
> Thank you for the valuable suggestions.
>
> This may be because using a return-prior aligns better with the distribution of the original trajectories. However, directly applying RTG-prior can introduce inconsistencies when reordering trajectories, as illustrated in Section 3.2-(2) of the revised version. Specifically, the resulting *rewards* (calculated as the difference between the RTGs of the current and next timesteps) in the re-ordered trajectories may sometimes lose important high-reward signals, since RTGs are extracted from different original trajectories and may no longer stay in high-reward regions.
>
> To illustrate this, consider a simple toy example with two original trajectories:
>
> Trajectory A: RTGs = [10, 2, 1] → rewards = [8, 1, 1]
>
> Trajectory B: RTGs = [7, 4, 3] → rewards = [3, 1, 3]
>
> If we use the return-prior, the reordering keeps each trajectory intact, preserving the original reward and RTG sequences.
>
> However, if we reorder based on RTG-prior, the transitions are sorted by RTG values across all data, potentially mixing transitions from different trajectories. For example, we might get:
>
> Reordered Traj 1: [10, 4, 3] → resulting rewards = [6, 1, 3]
>
> Reordered Traj 2: [7, 2, 1] → resulting rewards = [5, 1, 1]
>
> As a result, the highest original reward of 8 is no longer preserved in the reordered data. This demonstrates how RTG-prior can unintentionally discard valuable high-reward transitions, making return-prior a more stable option in some cases.
>
> -------
>
> (3) *(b) Why does DStitch suffer from more diverse datasets?*
>
> **Response**
>
> We did not observe that DStitch consistently suffers on more diverse datasets, as it generally improves performance over the Original evaluations across most settings. However, its improvements are often smaller compared to those achieved by our GODA method, particularly on diverse datasets. One possible explanation is that DStitch constructs new trajectories by generating stitching transitions between trajectories from the original dataset, failing to explore more possibilities beyond the original datasets. In contrast, GODA can generate transitions beyond the original dataset's scope. Besides, its core mechanism relies on smooth transitions between stitched segments. In highly diverse datasets, trajectories may span drastically different states, actions, or reward regimes (e.g., a dataset mixing successful and failed navigation paths in a complex environment). Stitching sub-trajectories from such diverse sources can create synthetic transitions that are physically implausible or semantically inconsistent (e.g., jumping from a "high-reward" state in one trajectory to a "low-reward" state in another without a valid action sequence connecting them). These limitations may contribute to DStitch's relatively weaker performance compared to GODA in diverse settings.

---

> ### Author Response · Authors · 2025-08-01
>
> (3) *(c) Why does sometimes Med-R perform better than Med, and other times the opposite trend is observed? For instance, Med-R is worse than Med for HC but better for WK2D (with IQL, Table 4). Do the authors have insights into which type of dataset is preferable and why?*
>
> **Response**
>
> Med-R datasets are collected from the full replay buffer of a moderately trained SAC policy. This means they contain a mix of early (suboptimal) and late (near-optimal) trajectories, leading to *higher diversity* in state-action distributions compared to Med (which uses only a partially trained policy’s rollouts, resulting in *higher-quality* demonstrations). Due to these differences in data collection methods, the performance of algorithms on the two datasets may vary. For example, some algorithms may be more suitable for the stable data distribution of the Medium dataset, while for other algorithms, the Medium-Replay dataset can provide more diverse and effective information, resulting in different performance of the same algorithm on these two datasets. Besides, different tasks have different complexity, which poses different difficulties for both the augmentation methods and evaluation algorithms. The failure of GODA on HP Med might be because of Walker2d's unstable dynamics and noisier data, which makes it harder for augmentation to yield consistent improvements.
>
> Ultimately, the preferable dataset depends on both the environment’s complexity and the algorithm’s robustness to diverse or imperfect data. However, from empirical results, our GODA works better on datasets with fewer optimal demonstrations (Random and Medium-Replay) compared with SynthER. This might be because that low-quality datasets have more room for RTG scaling while scaling on expert demonstration might sometimes result in unachievable data. Besides, our analysis on Section C.3 in Appendix shows that tasks with low reward sensitivity seem to be preferable for GODA.
>
> Moreover, from our experiments, we observe that GODA performs especially well in Maze2D navigation tasks, with a maximum improvement of 57.7% observed on the Maze2D-Large dataset. This is likely because the goals and states in Maze2D closely resemble the illustrative scenario shown in Figure 1. Maze2D tasks involve navigation from a start to a goal location, making the goal naturally explicit and spatially defined. GODA can directly condition the learning process on the desired goal position, which aligns perfectly with the task’s structure. Besides, GODA can possibly reuse sub-trajectories and recombine transitions from different paths that end near the same goal, making data augmentation more effective and less likely to violate task constraints.

---

> ### Author Response · Authors · 2025-08-01
>
> (4) *In Table 4, HP with Med-R has extremely different performance under IQL than under TD3+BC. Why is there such a dramatic gap between the two algorithms? For a RAND setting, I would have understood such a fluctuation, but what about the Med-R setting? Does that perhaps show that Med-R is inherently unstable?*
>
> **Response**
>
> Thank you for your thoughtful question. The large performance gap between IQL and TD3+BC on the Medium-Replay (Med-R) HalfCheetah dataset arises from both the nature of the dataset and the fundamental differences between the two algorithms.
>
> The Medium-Replay dataset contains all transitions collected during the training of a medium-performing agent. This includes a wide range of behaviors: early-stage, low-quality actions; noisy transitions; and gradually improving policy behaviors. Unlike static datasets such as Medium, which capture transitions from a fixed policy checkpoint, Med-R reflects the agent’s learning trajectory over time. This naturally introduces variability and noise into the dataset.
>
> IQL is robust to such suboptimal data. It performs policy updates using advantage-weighted regression, effectively ignoring low-advantage (i.e., poor-quality) actions. As a result, IQL can selectively learn from the higher-return parts of the dataset and largely disregard noisy or exploratory behaviors from earlier stages of training.
>
> In contrast, TD3+BC incorporates a behavior cloning term that encourages the policy to mimic all observed actions, regardless of quality. This uniform imitation can cause the policy to overfit to low-quality transitions present in Med-R, especially in early-stage data. Consequently, TD3+BC may suffer from degraded performance when trained on datasets with high behavioral variance, such as Med-R.
>
> This difference becomes more apparent after data augmentation with GODA. Since GODA augments high-return regions more effectively, algorithms like IQL, which can selectively exploit these improved samples, benefit more. TD3+BC, however, continues to treat all actions uniformly, limiting its ability to leverage the higher-quality transitions introduced by augmentation.
>
> Specifically, for HalfCheetah task, the agent can accumulate returns by continuously running forward, even with a suboptimal policy. However, the performance gap between mediocre and expert policies remains large. As a result, the augmented data further amplifies IQL’s advantage, while TD3+BC's inability to filter poor actions results in limited gains, hence the dramatic performance gap observed in Table 4.
>
> We have clarified this explanation in the revised manuscript to help readers better understand this discrepancy.

---

> ### Author Response · Authors · 2025-08-01
>
> (5) *I was wondering whether this framework could be used even in the context of online RL to get higher-quality data. The main difference would be that Algorithm 1 would be run periodically, and we would periodically re-train the goal-conditioned neural net. Have the authors considered such a use case?*
>
> **Response**
>
> Thank you for the insightful question. While our primary focus is on data augmentation for offline RL, as highlighted in the title and main body of our work, our framework can indeed be extended to the online RL setting, similar to the approach taken in SynthER.
>
> We show the pseudocode in **Algorithm 2 in the Appendix**. In the online extension, we periodically train the diffusion model using transitions collected from ongoing environmental interactions. The learned goal-conditioned diffusion model is then used to augment the dataset with synthetic transitions. The online policy is updated using a hybrid buffer containing both real and synthetic transitions in a 1:1 ratio.
>
> To evaluate this setting, we implemented our method using the Soft Actor-Critic (SAC) [1] algorithm and conducted experiments on three Gym tasks: HalfCheetah, Hopper, and Walker2D. For training: We train SAC with an Update-To-Data (UTD) ratio of 1 while train SAC (SynthER) and SAC (GODA) with a UTD of 20. For each 10K real data gathered, we enhance the dataset by augmenting 1M additional synthetic samples.
>
> Because the distribution and quality of the real buffer improve over time during online training, we adapted our hyperparameters accordingly. Specifically, we applied a decaying goal scaling factor, from 1.8 (effective in low-quality, random offline settings) to 1.1 (effective in expert-level offline settings), and an increasing Top-n condition selection, from 50 to 120, to reflect growing confidence in higher-quality goals as training progresses.
>
> We compare SAC (GODA) with both vanilla SAC and SAC (SynthER), and present online learning curves (**Figure 9 in the Appendix**) which shows evaluation returns over 100K interaction steps. As shown, GODA significantly outperforms the baselines in the early stages of training, where real transitions tend to be suboptimal. In the later stages, as the quality and quantity of real data improve, GODA continues to match or slightly outperform SAC (SynthER), with the performance gap gradually narrowing.
> These results demonstrate that GODA is not only effective in offline settings, but also a promising approach for enhancing online RL through adaptive, goal-conditioned data augmentation.
>
> [1] *Haarnoja, Tuomas, et al. "Soft actor-critic: Off-policy maximum entropy deep reinforcement learning with a stochastic actor." International conference on machine learning. Pmlr, 2018.*

---

> ### Author Response · Authors · 2025-08-01
>
> (6) *IQL and TD3+BC are not very recent methods. Have the authors considered more recent baselines? It would be interesting to compare the different settings under state-of-the-art offline RL methods.*
>
> **Response**
>
> Thank you for raising this important point. We agree that evaluating GODA with more recent offline RL algorithms is essential to demonstrate its generalizability.
>
> Following your suggestions, we have conducted experiments with several state-of-the-art offline RL methods, including Doubly Mild Generalization (DMG) [2], Conservative Policy Iteration (CPI) [3], and CPI with Reference Ensembles (CPI-RE) [3], all recently proposed in NeurIPS 2024 papers. Unfortunately, despite using the official open-source implementations, we were unable to reproduce most of the results reported for CPI and CPI-RE. As a result, we only present results for DMG, which are included in **Table 9 in Section C.2 in Appendix** of the revised version. DMG introduces a novel generalization strategy by selecting actions from a constrained neighborhood of the training data, thereby maximizing Q-values while avoiding excessive extrapolation. DMG also mitigates generalization propagation and aims to balance generalization and conservatism in offline RL.
>
> **Table: Performance comparison on Gym tasks using DMG as the evaluation algorithm. Values in bold represent the best performance (largest normalized score).**
>
> | Task           | Data           | Original     | SynthER      | GODA         |
> |----------------|----------------|--------------|--------------|--------------|
> | **HalfCheetah-v2** | Random         | 27.3±2.5     | 27.8±1.9     | **31.5±4.3**  |
> |                | Medium-Replay   | **51.4±1.6** | 47.1±2.4     | 50.9±1.7     |
> |                | Medium          | 55.2±1.8     | 57.0±0.6     | **60.5±1.1** |
> |                | Medium-Expert   | 41.3±2.9     | 44.2±11.3    | **46.2±10.6**|
> | **Walker2D-v2**    | Random         | 5.3±1.2      | 3.6±1.1      | **13.2±4.9** |
> |                | Medium-Replay   | 90.9±4.9     | 91.2±5.1     | **94.0±4.6** |
> |                | Medium          | 90.5±5.0     | **95.7±2.5** | 88.2±4.7     |
> |                | Medium-Expert   | 114.0±0.8    | **115.3±0.5**| 112.5±0.8    |
> | **Hopper-v2**      | Random         | 9.9±4.4      | 10.8±3.5     | **11.2±2.9** |
> |                | Medium-Replay   | **101.5±1.5**| 95.4±3.3     | 99.7±2.8     |
> |                | Medium          | 100.4±1.0    | **102.2±0.9**| **102.2±3.6**|
> |                | Medium-Expert   | 37.6±33.3    | **39.1±26.4**| 36.5±30.1    |
> | **Average**        |                | 60.4±5.1     | 60.8±5.0     | **62.2±6.0** |
>
>
> As shown in the table, GODA-enhanced DMG outperforms both the original DMG and the SynthER-enhanced DMG in 7 out of 12 evaluation scenarios, and achieves the highest average normalized score overall. It’s worth noting that we were unable to reproduce the results for HalfCheetah-Medium-Expert and Hopper-Medium-Expert for DMG. SynthER and our GODA also fail to improve the results of DMG on these two datasets to normal levels. We suspect this may be due to hyperparameter sensitivity in DMG rather than issues with the datasets themselves. Overall, these results support the claim that GODA can effectively generalize to more advanced offline RL algorithms and remains a promising data augmentation framework for enhancing state-of-the-art methods.
>
> [2] *Mao, Yixiu, et al. "Doubly mild generalization for offline reinforcement learning." Advances in Neural Information Processing Systems 37 (2024): 51436-51473.*
>
> [3] *Ma, Yi, et al. "Iteratively refined behavior regularization for offline reinforcement learning." Advances in Neural Information Processing Systems 37 (2024): 56215-56243.*

---

> ### Author Response · Authors · 2025-08-01
>
> (7) *It would have been interesting to see experiments with High-quality datasets, as opposed to Med and Med-R. This would allow one to see whether goal-conditioned data augmentation can raise Med or Med-R performance close to the optimum, or whether the gap remains. Ideally, we would like to reach a performance similar to a high-quality offline dataset after goal-conditioned data augmentation. Is that really what is happening? Insights into that could be very helpful.*
>
> **Response**
>
> Thank you for this valuable suggestion. We agree that incorporating high-quality datasets into our analysis is indeed critical to understanding whether goal-conditioned data augmentation can bridge the gap between medium-quality (Med/Med-R) and high-quality data.
>
> To address this, we have included additional experiments on Medium-Expert (Med-E) datasets, which consist of a 50/50 mixture of expert demonstrations and suboptimal trajectories. As shown in **Table 2** of the revised version, our results indicate that while GODA boosts performance on Med and Med-R datasets, a notable gap still remains between the performance of algorithms trained on GODA-augmented Med/Med-R datasets and the performance of algorithms trained on the original Med-Expert datasets.
>
> We believe this gap persists for several reasons: Expert demonstrations often contain fine-grained, task-specific patterns, such as precisely timed or finely tuned action sequences, that are difficult to reconstruct through augmentation alone. GODA operates by conditioning on high-return trajectories present in the dataset. When these trajectories are themselves suboptimal or sparse, the augmented data cannot fully replicate the quality and diversity found in true expert demonstrations. As a generative model, GODA may introduce slight inconsistencies or artifacts (e.g., unrealistic transitions or invalid states), which are rare in high-quality expert data.
>
> These limitations suggest that while GODA is effective at maximizing the utility of suboptimal data, it cannot fully synthesize expert-level performance from non-expert data alone. Its strength lies in extracting and amplifying high-quality signals within existing data, not inventing expert behavior from scratch. (This limitation has been added to Section 7.)
>
> Interestingly, we observe that GODA performs especially well in Maze2D navigation tasks, with a maximum improvement of 57.7% on the Maze2D-Large dataset. While GODA could not fully match expert-level performance, it significantly narrows the gap.
> We have added these insights to the revised version to clarify the capabilities and limitations of our method across different dataset quality levels and task types.

---

### Comment · Action_Editor_auVk · 2025-07-14
**Extension of author-reviewer discussion period**

Dear Authors and Reviewers,

Thank you for submitting and reviewing the manuscript.

There are additional experiments needed in the Requested Changes, where the authors find some more time helpful. The EiCs have extended the author-reviewer discussion period for 2 weeks (total 4 weeks). Please use this period to improve and discuss the manuscript.

Baoxiang

---

### Decision · Action_Editor_auVk · 2025-08-29

**Recommendation:** Accept as is

**Audience:**

Yes

**Audience Explanation:**

The work provides relevant insights to the community.

**Claims And Evidence:**

Yes

**Claims Explanation:**

This work introduces a goal-conditioned diffusion model to augment samples. The new samples will help the agent to learn in offline RL. Several techniques are proposed to accomplish this.

All reviewers and I find the contributions significant. The concerns are fixable, while the authors were able to fix them during the discussion period. The main claim is then supported.